# A Buffer Mechanism Employed in Language Models to Implement Symbolic Multi-Step Reasoning Tasks

## Abstract

Large language models have consistently struggled with complex reasoning tasks, such as mathematical problem-solving. Investigating the internal reasoning mechanisms of these models can help us design better model architectures and training strategies, ultimately enhancing their reasoning capability. In this study, we constructed a symbolic dataset to investigate the mechanisms by which Transformer models employ vertical thinking strategy based on their inherent structure and horizontal thinking strategy based on Chain of Thought to achieve multi-step reasoning. We introduced the concept of buffer mechanism: the model stores various information in distinct buffers and selectively extracts them through the query-key matrix. We proposed a random matrix-based algorithm to enhance the model's reasoning ability, resulting in a 75% reduction in the training time required for the GPT-2 model to achieve generalization capability on the PrOntoQA dataset. These findings provide new insights into understanding the mechanisms of large language models.

## 1 Introduction

In recent years, LLMs have emerged and demonstrated remarkable capability across various tasks (Vaswani et al., 2017; Liu et al., 2018; Devlin et al., 2018; Radford et al., 2019; Touvron et al., 2023; OpenAI, 2023). These models have shown impressive in-context learning abilities (Brown et al., 2020; Dong et al., 2022; Garg et al., 2022) and have been applied to logical reasoning problems, such as matching top human contestants at the International Mathematical Olympiad (IMO) level (Trinh et al., 2024) and solving math problems (Davies et al., 2021).

However, even the most advanced models still struggle with complex reasoning tasks, which indicates the ability of LLMs to handle multi-step logical reasoning remains constrained. To truly enhance the reasoning capability of LLMs, it is crucial to investigate their intrinsic mechanisms.

Multi-step reasoning tasks encompass a broad concept, typically referring to the ability of a model to synthesize numerous complex conditions to answer questions. Here, we consider a representative class of syntactic structures within multi-step reasoning tasks: a sentence that includes both the question and sufficient known information to answer it. For example, "Given: [A]→[B]...[B]→[C]..., Question: 2-step reasoning starting from [A]", where "..." represents other textual content unrelated to logical reasoning. The answer to this question is "[C]". When a sentence contains only one logical reasoning step, it is often handled by the so-called induction head in Transformer (Brown et al., 2020; Olsson et al., 2022). However, multi-step reasoning is not merely a linear accumulation of multiple induction heads but involves more complex mechanisms.

Large models employ two primary strategies for logical reasoning. The first, known as the *Vertical Thinking Strategy (VTS)*, outputs reasoning results in a single forward based on their inherent structure. This approach is efficient but demands a high level of intelligence from the model itself. As shown in Fig. 1, current large models exhibit significant limitations in their vertical thinking capability. Another relatively less efficient approach is the *Horizontal Thinking Strategy (HTS)*, such as Chain of Thought (CoT) (Wei et al., 2022; Kojima et al., 2022) and Tree of Thought (ToT) (Yao et al., 2024), Diagram of Thought (DoT) (Zhang et al., 2024a). This strategy substantially enhances the model's reasoning performance. All models can produce the correct answers for tasks depicted

**Question:** We have established the following reasoning rules: (1) [a] to [b] represents that condition [a] can derive condition [b]. (2) The sequence [a] to [b] | [b] to [c] indicates that the 2-step reasoning result of [a] is [c]. For the following reasoning chain:

[e] to [i] | [r] to [w] | [n] to [a] | [o] to [p] | [i] to [r] | [p] to [e] | [w] to [p] | [x] to [i]

Please answer directly: What is the ****-step reasoning result of [w]? (only return the answer)

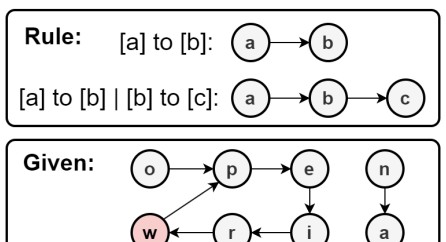

**Rule:**   [a] to [b]:

[a] to [b] | [b] to [c]:

**Given:**

**Accuracy:**

| | GPT4 | GPT4o | Claude3.5 haiku | Clauld3.5 sonnet |
|---|---|---|---|---|
| 2-step | 52% | 44% | 24% | 100% |
| 3-step | 0% | 4% | 0% | 4% |
| 4-step | 0% | 4% | 4% | 12% |

Figure 1: The interaction results of multi-step reasoning tasks with large models. We tested each model 25 times, and when the number of reasoning steps exceeded three, these models exhibited random guessing. However, when we allowed the models to use CoT prompting (by removing "directly" and "only return the answer"), all models achieved 100% accuracy. Detailed interaction results are provided in the Appendix I.

in Fig. 1 with the help of CoT. Considering the strengths and weaknesses of these two strategies, the ideal approach should combine both: decomposing problems into several coarse-grained subproblems (HTS) and applying the VTS to each subproblem. Thus, researching how to improve vertical thinking capability and understanding why horizontal thinking strategy can significantly enhance model reasoning ability are both crucial.

In this work, we investigate the performance of Transformer models on a symbolic multi-step reasoning dataset. Our work aims to uncover these mechanisms and provide insights into how Transformer processes and integrates logical information across multiple layers to perform multi-step reasoning, which can help develop more effective strategies for improving their multi-step reasoning abilities. Specifically, we found that Transformers utilize a ***Buffer Mechanism*** when engaging in symbolic multi-step reasoning tasks. The model stores different intermediate results in separate buffers, allowing for quick retrieval as needed. We elaborate on how the model leverages this buffer mechanism for vertical thinking and we explain why horizontal thinking strategy can significantly enhance the model's multi-step reasoning capability from the perspective of the buffer mechanism. Finally, based on this understanding, we propose a method to enhance the model's reasoning abilities, resulting in a 75% reduction in the cost of generalization for the GPT-2 model on the question-answering dataset PrOntoQA (Saparov & He, 2022). The concept of "buffer" or similar concepts has also been mentioned in other works (Reddy, 2023; Bietti et al., 2024; Elhage et al., 2021). Our work provides a detailed description of the concept of buffer and applies this mechanism to enhance model performance.

The main contributions of this work are as follows:

- We propose a buffer mechanism and found evidence that supports such mechanism being employed by language models during the reasoning process in symbolic multi-step reasoning tasks and provide a detailed analysis of the model's internal thinking process for vertical and horizontal thinking from the perspective of the buffer.

- We propose a method to enhance the model's reasoning capability, significantly improving data utilization efficiency in logical reasoning datasets.

Our research deepens the understanding of the reasoning mechanisms in Transformer models and provides new perspectives for further enhancing their reasoning capability. The insights gained from this study can contribute to the design of more efficient reasoning models and the exploration of reasoning mechanisms in general artificial intelligence systems.

## 2   REASONING DATASET AND TRANSFORMER MODEL

**Dataset.** To understand the mechanism of multi-step reasoning in Transformers, we design an abstract symbolic multi-step reasoning task. As shown in Fig. 2, reasoning chains are serialized into a sequence. Every two tokens in the sentence represent a reasoning relation. The last token is the reasoning start token, and the label is the reasoning result with a fixed-step reasoning starting from the starting point.

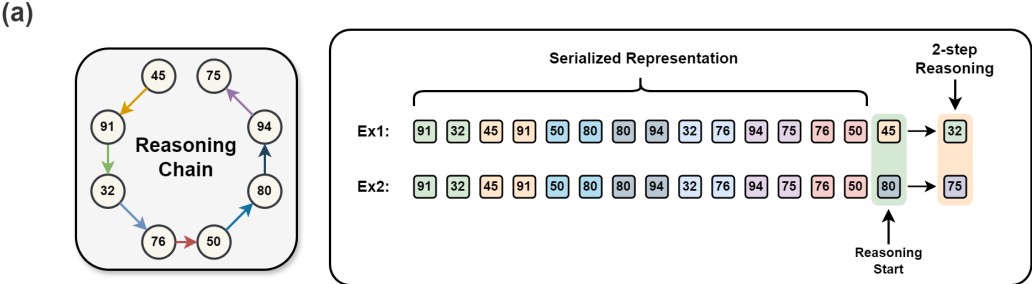

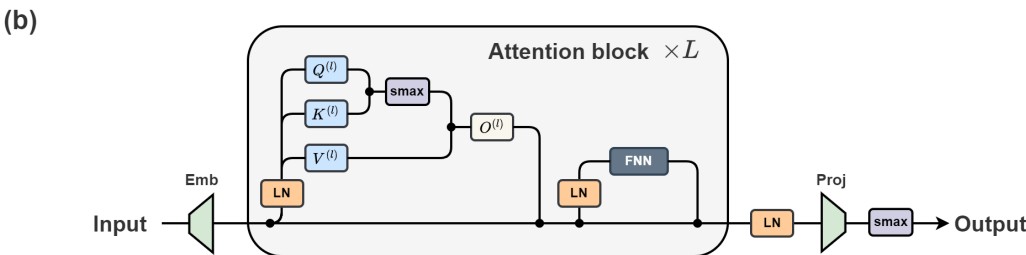

Figure 2: Illustration of the dataset and Transformer structure. (a) We investigate reasoning chains composed of digital tokens. In a serialized representation, each pair of adjacent tokens (represented by the same color) forms a reasoning relation within the reasoning chain. The order of the reasoning relations is random, thus a single reasoning chain can correspond to various serialized representations. The model input consists of the serialized representation along with a reasoning start token. The label is the result of performing a fixed number of reasoning steps starting from this start token. This figure illustrates a 2-step reasoning task. (b) We employ a standard decoder-only Transformer architecture for training.

**Transformer Model.** We employ a decoder-only Transformer. Given an input sequence $\boldsymbol{X}^{\text{in}} \in \mathbb{R}^{n \times d}$, where $n$ is the sequence length and $d$ is the dictionary size, the model first applies an embedding layer (target embedding and position embedding) to obtain the input representation $\boldsymbol{X}^{(1)} = \boldsymbol{X}_{tgt} + \boldsymbol{X}_{pos} \in \mathbb{R}^{n \times d_m}$. The single-head attention in each layer is computed as follows:

$$\mathcal{A}^{(l)}(\boldsymbol{X}) = \sigma\left(\frac{\text{mask}(\boldsymbol{X}\boldsymbol{W}^{q(l)}\boldsymbol{W}^{k(l),\mathsf{T}}\boldsymbol{X}^{\mathsf{T}})}{\sqrt{d_k}}\right), \quad \boldsymbol{X}^{\text{qkv}(l)} = \mathcal{A}^{(l)}(\bar{\boldsymbol{X}}^{(l)})\bar{\boldsymbol{X}}^{(l)}\boldsymbol{W}^{v(l)}\boldsymbol{W}^{o(l)},$$

where $\sigma$ takes the softmax operator, and $\bar{\boldsymbol{X}}^{(l)} = \text{Layernorm}(\boldsymbol{X}^{(l)})$. For simplicity of expression, we will abbreviate $\boldsymbol{W}^{q(l)}\boldsymbol{W}^{k(l),\mathsf{T}}$ as $\boldsymbol{W}^{qk(l)}$ and $\boldsymbol{W}^{v(l)}\boldsymbol{W}^{o(l),\mathsf{T}}$ as $\boldsymbol{W}^{vo(l)}$ in the following text. The output of the $l$-th layer is obtained as:

$$\boldsymbol{X}^{\text{ao}(l)} = \boldsymbol{X}^{(l)} + \boldsymbol{X}^{\text{qkv}(l)}, \quad \boldsymbol{X}^{(l+1)} = f^{(l)}(\bar{\boldsymbol{X}}^{\text{ao}(l)}) + \boldsymbol{X}^{\text{ao}(l)},$$

where $f^{(l)}(\cdot)$ represents the feedforward neural network of $l$-th layer. The final output (in the form of token indices within the vocabulary) is obtained as:

$$\boldsymbol{Y} = argmax(\sigma(\bar{\boldsymbol{X}}^{(L)}\boldsymbol{W}^p)) \in \mathbb{R}^n. \tag{1}$$

**In Distribution and Out of Distribution Data.** With the settings of our dataset, if the model truly understands the underlying logic patterns, it should be able to find the correct answer to the sentence,

even if this sentence has tokens that have never been encountered during the training. Therefore, we divided the data into two parts: in-distribution (ID) and out-of-distribution (OOD). Specifically, we define $\texttt{token}_{\text{ID}} \in [20, 100]$ and $\texttt{token}_{\text{OOD}} \in [0, 120] \setminus [20, 100]$. In-distribution data ($\texttt{Train}_{\text{ID}}$ and $\texttt{Test}_{\text{ID}}$) is defined as sentences composed entirely of $\texttt{token}_{\text{ID}}$, while out-of-distribution data ($\texttt{Test}_{\text{OOD}}$) consists of sentences containing one $\texttt{token}_{\text{OOD}}$, which happens to be the previous reasoning step of label. For the in-distribution data, we split the training set ($\texttt{Train}_{\text{ID}}$) and test set ($\texttt{Test}_{\text{ID}}$) according to the following rules: for the serialized reasoning chain $[\texttt{x}_1][\texttt{x}_2]\cdots[\texttt{x}_n]$ of the training set, all tokens satisfy the following condition:

$$\texttt{x}_{2\texttt{i}} - \texttt{x}_{2\texttt{i-1}} \pmod{m} \in G. \tag{2}$$

For the reasoning chains in the test set, all tokens satisfy:

$$\texttt{x}_{2\texttt{i}} - \texttt{x}_{2\texttt{i-1}} \pmod{m} \in \{1, \cdots, m\}\setminus G, \tag{3}$$

where we take $m = 5$ and $G = \{0, 1, 4\}$ in this study. Under this setting, we ensure that each binary logical pair in the testing set has not previously appeared in the training set. Therefore the Transformer is performing in-context learning (Brown et al., 2020; Olsson et al., 2022), as each reasoning pair is not seen during in-weight learning.

## 3 BUFFER MECHANISM AND VERTICAL THINKING STRATEGY

In this section, we examine the mechanism by which the Transformer model employs a vertical thinking strategy for multi-step reasoning. Through conducting causal intervention(Feng & Steinhardt, 2023; Meng et al., 2022; Vig et al., 2020; Wang et al., 2024a) (Appendix J), we discovered that when the number of layers exceeds the number of reasoning steps, the Transformer's logic circuit of vertical thinking can be depicted in Fig. 3. The first layer facilitates the fusion of token information at odd and even positions. In each subsequent layer, which we refer to as the reasoning layer, the token at the last position attends to the token carrying the next reasoning result based on the existing reasoning result, thereby accomplishing one-step reasoning.

### 3.1 BUFFER MECHANISM

It is important to note the distinction between information transmission triggered by attention and that induced by residual connections. In the attention mechanism, information $[\texttt{a}]$ with dimension $d_m$ is first mapped to a lower embedding dimension $d_v$ through the matrix $\boldsymbol{W}^v$, and then it is projected back to $d_m$ dimensions via the matrix $\boldsymbol{W}^o$. Consequently, the information $[\texttt{a}]$ after passing through the attention layer becomes $[\texttt{a}]\boldsymbol{W}^{vo}$. In contrast, information transmitted through residual connections does not require additional processing; thus, after the first layer, the information at even indices transforms to $[\texttt{x}_{2\texttt{i-1}}]\boldsymbol{W}^{vo(0)} + [\texttt{x}_{2\texttt{i}}]$.

We observe that each token in Fig. 3 stores various information in various forms (projected by different matrices $\boldsymbol{W}^{vo(l)}$), which leads to a natural question: how does the attention layer effectively retrieve useful information while avoiding interference from others? To address this question, we introduce the following lemma (the proof can be found in Appendix C):

**Lemma 1.** *Suppose token $\boldsymbol{x} = \sum_{i=1}^{n} \boldsymbol{a}_i \boldsymbol{W}_i \in \mathbb{R}^{d_m}$ and token $\boldsymbol{y} = \sum_{i=1}^{n} \boldsymbol{b}_i \boldsymbol{W}_i \in \mathbb{R}^{d_m}$, where $\boldsymbol{a}_i, \boldsymbol{b}_i \in \mathbb{R}^{d_m}$, $\boldsymbol{W}_i \in \mathbb{R}^{d_m \times d_m}$, $i = 1, 2, \cdots, n$. Each element of $\{\boldsymbol{a}_i\}_{i=1}^{n}$, $\{\boldsymbol{b}_i\}_{i=1}^{n}$ and $\{\boldsymbol{W}_i\}_{i=1}^{n}$ follows a normal distribution $\mathcal{N}(0, 1/d_m)$ and independent to others. Then, we have:*

$$\boldsymbol{x}\boldsymbol{W}_i^{\mathsf{T}} = \boldsymbol{a}_i + \mathcal{O}\left(\sqrt{\frac{n}{d_m}}\right), \quad \boldsymbol{y}\boldsymbol{W}_j^{\mathsf{T}} = \boldsymbol{b}_j + \mathcal{O}\left(\sqrt{\frac{n}{d_m}}\right), \tag{4}$$

$$\boldsymbol{x}\boldsymbol{W}_i^{\mathsf{T}}\boldsymbol{W}_j\boldsymbol{y}^{\mathsf{T}} = \boldsymbol{a}_i\boldsymbol{b}_j^{\mathsf{T}} + \mathcal{O}\left(\frac{n}{\sqrt{d_m}}\right). \tag{5}$$

It can be observed that the matrices $\{\boldsymbol{W}_i\}_{i=1}^{n}$ serve as a set of **buffers** for information. Each element of $\{\boldsymbol{a}_i\}_{i=1}^{n}$ is located in different buffers, and is almost unaffected by others. This property also applies for $\{\boldsymbol{b}_i\}_{i=1}^{n}$. By selecting the matrices associated with the relevant buffer, specific information contained in token $\boldsymbol{x}$ and token $\boldsymbol{y}$ can be extracted.

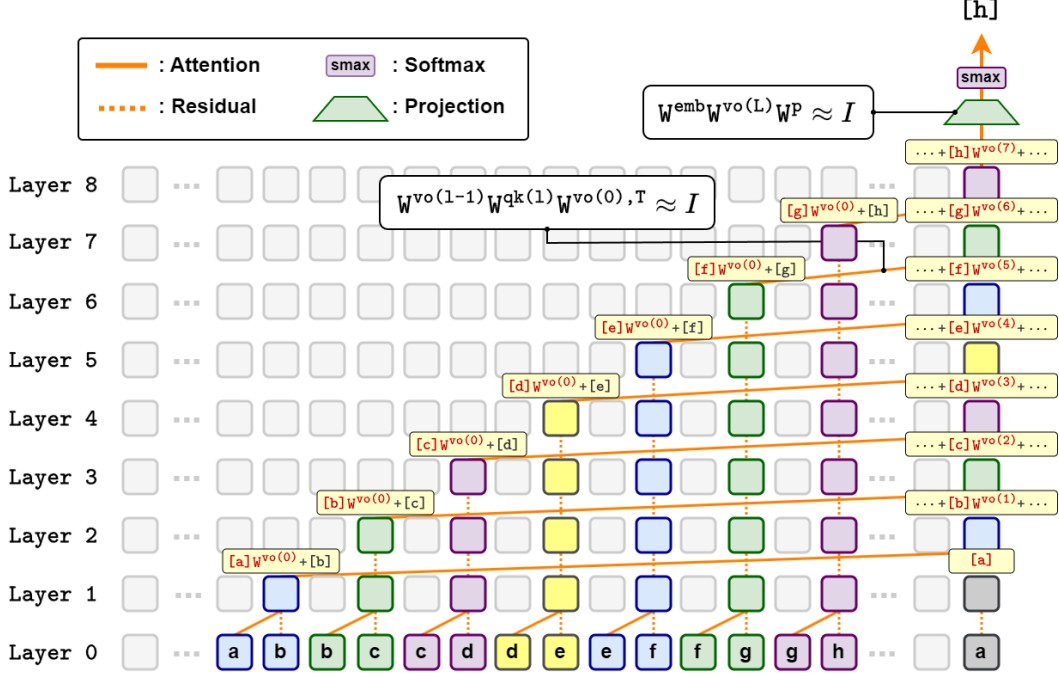

Figure 3: Illustration of how the Transformer model employs vertical thinking strategy for multi-step reasoning task. The first layer of attention pairs tokens at odd positions with those at even positions. Subsequently, each layer passes a new reasoning result to the last token. We annotate the information carried by the important token. The information crucial for the layer's information transfer is highlighted in red text.

The concept of the buffer mechanism is useful for understanding the internal logic mechanisms of Transformer models. $\{\boldsymbol{W}^{q(l)}\}_{l=1}^{L}$ and $\{\boldsymbol{W}^{k(l)}\}_{l=1}^{L}$ can be viewed as "information extractors", while $\{\boldsymbol{W}^{vo(l)}\}_{l=1}^{L}$ can be considered as a set of buffers. We observe that in the final token of each reasoning layer, each new intermediate result is always associated with a new $\boldsymbol{W}^{vo}$. If these $\{\boldsymbol{W}^{vo(l)}\}_{l=1}^{L}$ matrices are mutually orthogonal or random, then these intermediate results can be regarded as being stored in a new buffer.

In each reasoning layer, the token at the last position contains the current intermediate result. Additionally, there exists a token that contains both the current intermediate result and the next step's reasoning result, with these stored in different buffers. The role of $\boldsymbol{W}^q$ and $\boldsymbol{W}^k$ is to extract the current intermediate results from these two tokens, enabling them to attend to each other due to having the same token. We refer to this feature as "*same-token matching*". To quantitatively characterize this property, we define the following match matrix:

$$h^{(1)}(\boldsymbol{X}) = \boldsymbol{X}\boldsymbol{W}^{qk(1)}\boldsymbol{W}^{vo(0),\mathsf{T}}\boldsymbol{X}^{\mathsf{T}} \triangleq \boldsymbol{X}\,\mathrm{Ker}^{(1)}\,\boldsymbol{X}^{\mathsf{T}}, \tag{6}$$

$$h^{(l)}(\boldsymbol{X}) = (\boldsymbol{X}\boldsymbol{W}^{vo(l-1)})\boldsymbol{W}^{q(l)}\boldsymbol{W}^{k(l),\mathsf{T}}(\boldsymbol{X}\boldsymbol{W}^{vo(0)})^{\mathsf{T}} \tag{7}$$

$$= \boldsymbol{X}\boldsymbol{W}^{vo(l-1)}\boldsymbol{W}^{qk(l)}\boldsymbol{W}^{vo(0),\mathsf{T}}\boldsymbol{X}^{\mathsf{T}} \triangleq \boldsymbol{X}\,\mathrm{Ker}^{(l)}\,\boldsymbol{X}^{\mathsf{T}}, \, l \geq 2. \tag{8}$$

where

$$\mathrm{Ker}^{(1)} = \boldsymbol{W}^{qk(1)}\boldsymbol{W}^{vo(0),\mathsf{T}}, \tag{9}$$

$$\mathrm{Ker}^{(l)} = \boldsymbol{W}^{vo(l-1)}\boldsymbol{W}^{qk(l)}\boldsymbol{W}^{vo(0),\mathsf{T}}, \, l \geq 2. \tag{10}$$

We temporarily ignore the effects introduced by the feedforward layers only in our analysis; a detailed version that includes the feedforward layers can be found in Appendix D.

To achieve the same-token matching, it is sufficient for $\mathrm{Ker}^{(l)} \approx I$, in which case

$$h^{(l)}(\boldsymbol{X}_{tgt}) \approx \boldsymbol{X}_{tgt}\boldsymbol{X}_{tgt}^{\mathsf{T}} = \boldsymbol{I} + \mathcal{O}\left(\frac{1}{\sqrt{d_m}}\right). \tag{11}$$

Eq.( 11) means that the attention of all tokens is focused on themselves.

Furthermore, a much more remarkable observation is that the same-token matching is independent of the specific value of $\boldsymbol{X}_{tgt}$. For example, for $\boldsymbol{X}_{tgt,\text{OOD}}$ sampled from the untrained random vectors $\text{token}_{\text{OOD}}$, $h^{(l)}(\boldsymbol{X}_{tgt,\text{OOD}}) \approx \boldsymbol{I}$ still holds. Therefore, when the model's weights satisfy $\text{Ker}^{(l)} \approx I$, the model exhibits out-of-distribution generalization capability.

### 3.2 EXPERIMENTS

We utilized a 3-layer, single-head Transformer model to learn from a two-step reasoning dataset. Specific Experimental settings can be found in Appendix B. After training, the Transformer exhibits generalization capability in in-distribution (100% accuracy) and out-of-distribution data (over 82% accuracy).

Fig. 4 presents the experimental validation of Eq.(9) $\sim$ Eq.(11). We visualize the computed values of $h^{(1)}(\boldsymbol{X}_{tgt})$, $h^{(2)}(\boldsymbol{X}_{tgt})$, $\text{Ker}^{(1)}$, and $\text{Ker}^{(2)}$ with the model parameters after training. Notably, in the $\text{token}_{\text{OOD}}$ region, $\text{token}_{\text{OOD}}$ and $h^{(2)}(\boldsymbol{X}_{tgt})$ exhibit a diagonal structure similar to an identity matrix $\boldsymbol{I}$, ensuring the model's OOD generalization capability.

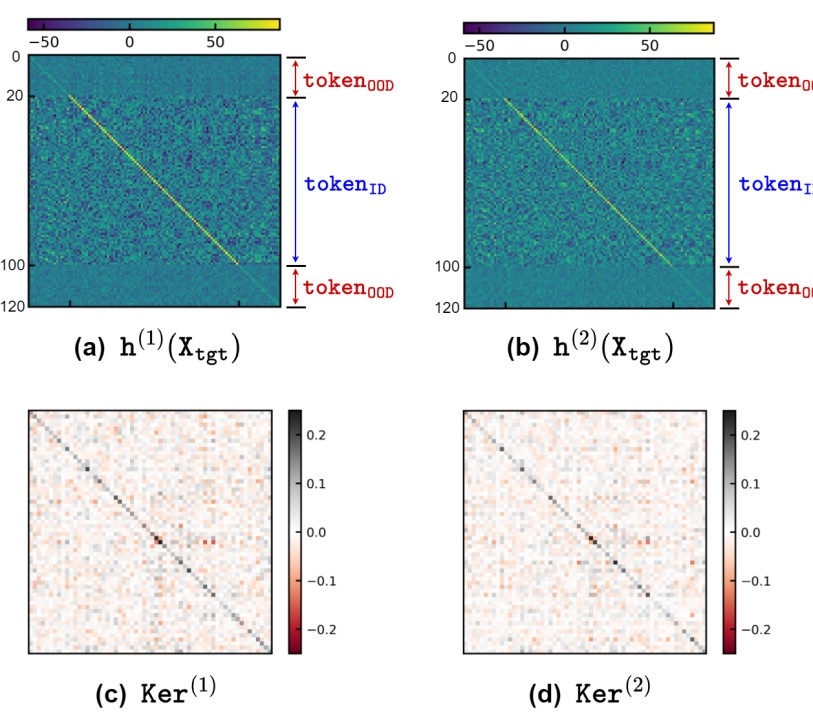

Figure 4: (a) (b) Heatmap of $h^{(1)}(\boldsymbol{X}_{tgt})$ and $h^{(2)}(\boldsymbol{X}_{tgt})$. (c) (d) Heatmap of $\text{Ker}^{(1)}$ and $\text{Ker}^{(2)}$. For better presentation, we only plot $\text{Ker}^{(1,2)}_{[:60,:60]}$ (the actual dimension of $\text{Ker}^{(1,2)}$ is $400 \times 400$). According to (c)(d), and Eq.( 11), the diagonal structure of the kernel matrix induces a diagonal structure in the matching matrix, even when $\boldsymbol{X}_{tgt}$ is sampled from the $\text{token}_{\text{OOD}}$.

To establish a strong correlation between the model's OOD generalization and its ability to use the buffer mechanism for same-token matching, we define a metric for this capability, the matching score:

$$\text{Matching Score}(h^{(l)}) = \mathbb{E}_{\boldsymbol{X}}[\text{Trace}(\sigma(h^{(l)}(\boldsymbol{X})))]/n, \tag{12}$$

where $\boldsymbol{X} \in \mathbb{R}^{n \times d_m}$ is sampled from $\text{token}_{\text{ID}}$ or $\text{token}_{\text{OOD}}$ for expectation $\mathbb{E}_{\boldsymbol{X}}$, and $\sigma$ takes the softmax operation. The level of diagonalization of $\text{Ker}^{(l)}$ serves as the intrinsic driver for achieving same-token matching; thus, we also define a metric for this diagonalization level, the kernel score:

$$\text{Kernel Score}(\text{Ker}^{(l)}) = \text{Trace}(\sigma(\text{Ker}^{(l)}))/d_m \tag{13}$$

Fig. 5 illustrates the accuracy curve and the dynamic changes in the matching score and kernel score during training. It is observed that the increase in the model's ID and OOD generalization coincide with the increases in the model's matching score and kernel score. By computing the cosine similarity between different buffers ($\boldsymbol{W}^{vo(0)}, \boldsymbol{W}^{vo(1)}, \boldsymbol{W}^{vo(2)}$ and $I$), we observe that these buffers are nearly pairwise orthogonal(Appendix D). Based on the above experimental analysis, we conclude that the Transformer model indeed employs the buffer mechanism for vertical thinking.

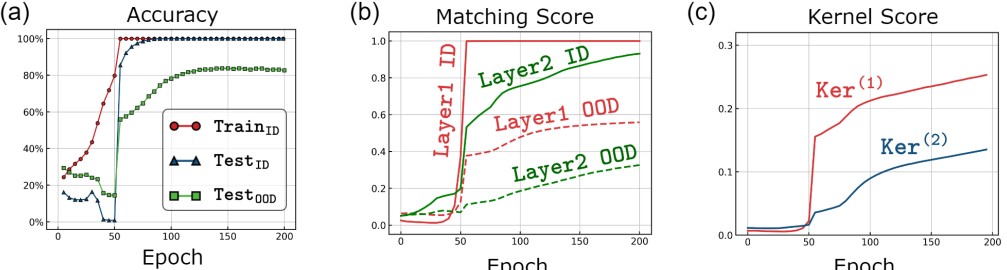

Figure 5: (a) Accuracy curve during training. (b) The dynamic evolution of the model's matching score. The red and blue lines represent the matching scores for the first and second layers, respectively. Solid and dashed lines indicate the matching scores when $\boldsymbol{X}$ is drawn from token$_{\text{ID}}$ and token$_{\text{OOD}}$, respectively. (c) The kernel scores for the first (red) and second (blue) layers.

According to the Buffer mechanism, in theory, by setting the model's weights in the following way, an $L$-layer model can achieve $(L-1)$-step reasoning *without the extra training process*:

$$\boldsymbol{W}^{q(0)} = \boldsymbol{W}^{q(1)} = I, \ \boldsymbol{W}^{q(l)} = \boldsymbol{W}^{vo(l-1),\mathsf{T}}, \ l \geq 2, \tag{14}$$

$$\boldsymbol{W}^{k(0)} = \sum_{i=1}^{[l_{\text{seq}}/2]} \boldsymbol{p}_{2i} \boldsymbol{p}_{2i-1}^{\mathsf{T}}, \ \boldsymbol{W}^{k(l)} = \boldsymbol{W}^{vo(0),\mathsf{T}}, \ l \geq 1, \tag{15}$$

where $\{\boldsymbol{W}^{vo(l)}\}_{l=1}^{L}$ are set as random matrices and the projection layer satisfies $\boldsymbol{W}^p = \boldsymbol{W}^{vo(L),\mathsf{T}} \boldsymbol{W}^{\text{emb},\mathsf{T}}$. Under this configuration, an 8-layer Transformer model can achieve 7-step reasoning (See Appendix D).

Another natural question is whether the model can still employ this mode of reasoning when the number of layers $L$ is less than the number of reasoning steps. In fact, in this scenario, the model exhibits the ability to engage in parallel thinking. A possible explanation is that the model can partition $2^L$ buffers through $\{\prod_{l \in J} \boldsymbol{W}^{vo(l)} | J \subset \{0, 1, \cdots, L-1\}\}$ since each $\prod_{l \in J} \boldsymbol{W}^{vo(l)}$ can be regarded as a new random matrix with the same distribution as $\boldsymbol{W}^{vo}$. We discuss this issue in detail in Appendix E.

In Appendix H, we provide additional definitions for methods to compute the matching score and kernel score in multi-head models and perform these calculations in the large language model Phi-3. Similar phenomena, such as weight alignment and $\boldsymbol{W}^{vo}$ diversity provide evidence for the presence of the buffer mechanism in real language models.

## 4 UNDERSTANDING HORIZONTAL THINKING STRATEGY WITH BUFFER MECHANISM

The Chain of Thought (CoT) approach, as a representative of horizontal thinking strategy, has become a prevalent mode of reasoning adopted by current large models, with numerous studies indicating that CoT can significantly enhance logical reasoning capability (Wei et al., 2022; Kojima et al., 2022). In this section, we will demonstrate that a Transformer model utilizing CoT can achieve arbitrary multi-step reasoning with only two layers. We trained a 2-layer Transformer with the 13-length single-step reasoning data. During the testing phase, we fed the model's output back into the model. Through this CoT process, the model can perform 2-step, 3-step, or even higher-step reasoning and it can also generalize to sentence lengths beyond the 13th position. The specific information

flow can be found in Appendix G. Fig. 6 presents a schematic diagram illustrating how the model implements this process. In the previously mentioned vertical thinking scenario, the model needed to allocate new buffer $\boldsymbol{W}^{vo(1)}$, $\boldsymbol{W}^{vo(2)}$ for storing the intermediate result [b] and [c], respectively. In contrast, with CoT, the model generates a new token to separately store the new intermediate information, effectively replacing the information [a] in the original buffer $\boldsymbol{W}^{vo(1)}$, $\boldsymbol{W}^{vo(0)}$ and $\boldsymbol{I}$ (identity matrix, which can also be treated as a buffer). Thus, CoT performs multi-step reasoning tasks through buffer reuse. Unlike the vertical thinking strategy, which requires alignment across multiple weight matrices, the horizontal thinking strategy only requires a few layers to satisfy $\text{Ker}^{(l)} \approx I$. This significantly reduces the difficulty of model training. This may also explain why CoT strategy can significantly enhance the reasoning capabilities of real large language models.

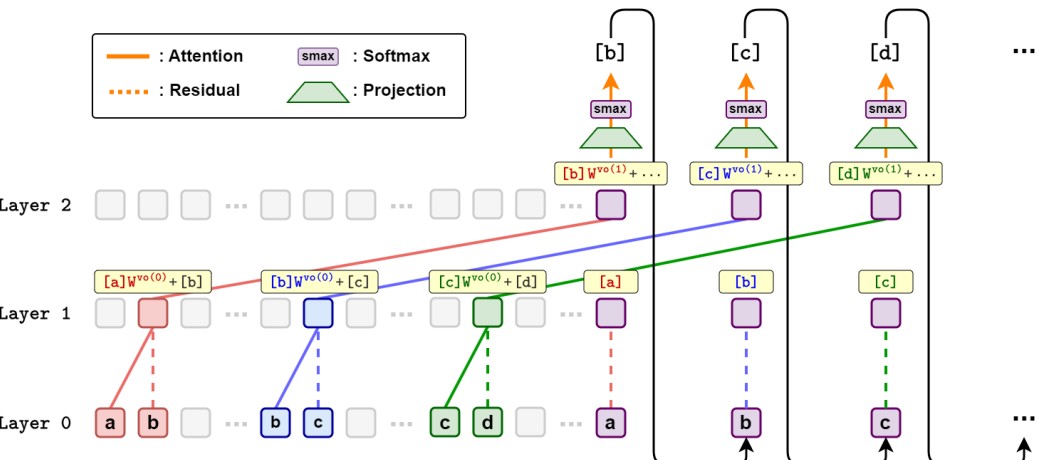

Figure 6: Illustration of how the Transformer model employs horizontal thinking strategy for multi-step reasoning task. The red, blue, and green portions indicate the information activated during the first, second, and third reasoning steps, respectively. Unlike vertical thinking, during horizontal thinking, the model repeatedly writes and retrieves new information from the buffers $\boldsymbol{W}^{vo(1)}$, $\boldsymbol{W}^{vo(0)}$, and $\boldsymbol{I}$ to perform multi-step reasoning. For example, the information [b] in Layer 2 is initially stored in the buffer $\boldsymbol{W}^{vo(1)}$, but when it is reintroduced into the model as a new token, it is transferred to the buffer $\boldsymbol{I}$. The next-step reasoning result [c] is then written into the buffer $\boldsymbol{W}^{vo(1)}$ in Layer 2.

## 5 METHOD FOR IMPROVING TRANSFORMER'S DATA-EFFICIENCY

In this section, we discuss how to improve data efficiency when employing the **vertical thinking strategy**, specifically, by enabling the model to learn the essence of logical reasoning with less data or training epoch. As mentioned in Section 3, this can be achieved by setting $\boldsymbol{W}^{vo(l-1),\mathsf{T}}\boldsymbol{W}^{qk(l)}\boldsymbol{W}^{vo(0),\mathsf{T}} \approx \boldsymbol{I}$.

A more intuitive approach is to replace $\boldsymbol{W}^{qk(l)}$ and $\boldsymbol{W}^{vo(l)}$ with $\boldsymbol{W}^{qk(l)}+\alpha^{(l)}I$ and $\boldsymbol{W}^{vo(l)}+\beta^{(l)}I$, where $\{\alpha^{(l)}\}_{l=1}^{L}$ and $\{\beta^{(l)}\}_{l=1}^{L}$ are learnable parameters. This Identity Matrix-Based Algorithm (denoted as IMBA) was first proposed in Boix-Adsera et al. (2023) and has been validated from both theoretical and empirical perspectives for its role in facilitating the model's learning of one-step reasoning data. However, in multi-step reasoning tasks, IMBA may lead to information interference. For instance, if $\boldsymbol{W}^{vo(l)}$ is replaced with $\boldsymbol{W}^{vo(l)} + \beta^{(l)}\boldsymbol{I}$, the storage representation of the two pieces of information transitions from [a]$\boldsymbol{W}^{vo}$+[b] to [a]$\boldsymbol{W}^{vo}$+($\beta^{(l)}$[a]+[b]), which introduces interference among the different pieces of information. Therefore, this approach may not be effective in enhancing the model's ability to perform multi-step reasoning.

Based on the understanding of buffer mechanism, we propose a **Random Matrix-Based Algorithm (RMBA)**, specifically by substituting $\boldsymbol{W}^{qk(l)}$ and $\boldsymbol{W}^{vo(l)}$ with $\boldsymbol{W}^{qk(l)}+\alpha^{(l)}\boldsymbol{Z}^{(l-1)}$ and $\boldsymbol{W}^{vo(l)}+\beta^{(l)}\boldsymbol{Z}^{(l)}$, where $\{\alpha^{(l)}\}_{l=1}^{L}$ and $\{\beta^{(l)}\}_{l=1}^{L}$ are learnable parameters and $\{\boldsymbol{Z}^{(l)}\}_{l=1}^{L}$ is a set of fixed

random matrix following $N(0, 1/d_m)$. In this case, the information storage representation changes from [a]$\boldsymbol{W}^{vo}$+[b] to [a]$\boldsymbol{W}^{vo}$+[a]$\boldsymbol{Z}$+[b], effectively creating a new buffer.

Fig. 7 illustrates the different results of the two methods. The baseline is a three-layer transformer model, which fails to learn a two-step reasoning task with only 30,000 samples. Under various hyperparameter settings, when $\alpha_{\text{ini}}^{(l)} \cdot \beta_{\text{ini}}^{(l)} \geq 0$, the RMBA algorithm significantly enhances the model's generalization ability, while the IMBA shows no effect. This experiment validates our buffer mechanism understanding. To strengthen the credibility of our results, we conducted a comprehensive sweep of hyperparameters such as weight decay, learning rate, and hidden dimensions ($d_m$ and $d_k$). The findings indicate that RMBA parameterization can more robustly reach a high accuracy by a certain number of training steps across a wider range of hyperparameters. Detailed parameter settings and experimental results can be found in Appendix F.

These results also demonstrate that multi-step reasoning is not achieved by simply "stacking" multiple single-step reasonings. An algorithm that enhances single-step reasoning may not be applicable to multi-step reasoning. Therefore, investigating the mechanisms of multi-step reasoning is an important and meaningful topic.

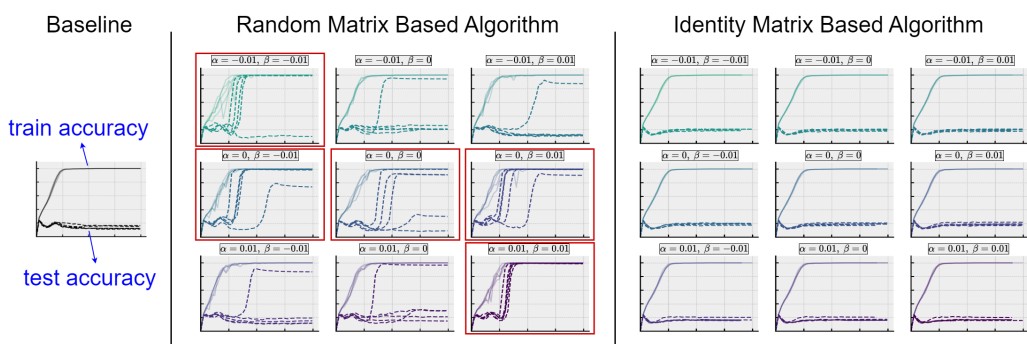

Figure 7: The accuracy comparison for the Baseline model, RMBA model, and IMBA model. The solid lines represent the training accuracy, while the dashed lines denote the test accuracy. When the training data is limited (only 30,000 samples), the Baseline model lacks generalization capability. For both RMBA and IMBA models, we set 9 different initialization parameter values $\alpha_{\text{ini}}^{(l)}$ and $\beta_{\text{ini}}^{(l)}$ and each experiment was conducted with 5 random seeds (each seed corresponds to one line). When $\alpha_{\text{ini}}^{(l)} = 0$ or $\alpha_{\text{ini}}^{(l)} \cdot \beta_{\text{ini}}^{(l)} > 0$, the model's reasoning capability can be enhanced. Detailed settings can be found in Appendix F.

## 5.1 VALIDATION OF ALGORITHM PERFORMANCE ON PRONTOQA DATASET

PrOntoQA dataset (Saparov & He, 2022) is constructed to test the reasoning capability of language models. We use the data generation procedure to create a 0.5M dataset of 3-step reasoning question-answer pairs, formatted as shown in Fig. 8(a).

We use a 12-layer, 12-head GPT-2 small model for this dataset. For the IMBA, we followed the settings provided in the open-source code (Boix-Adsera et al., 2023), which includes 144 learnable parameters $\alpha^{(l,h)}$ and 144 learnable parameters $\beta^{(l,h)}$. For RMBA, we replace $\boldsymbol{W}^{qk(l,h)}$ and $\boldsymbol{W}^{vo(l,h)}$ with $\boldsymbol{W}^{qk(l,h)} + \alpha^{(l,h)}\boldsymbol{Z}^{(l-1,h)}$ and $\boldsymbol{W}^{vo(l,h)} + \beta^{(l,h)}\boldsymbol{Z}^{(l,h)}$. Fig. 8(b) illustrates the impact of different algorithms on the performance of the original GPT-2 small model (baseline) on this dataset. In this figure, accuracy reflects the model's precision in predicting True or False for the last token. The GPT2-baseline model begins to generalize after approximately 1 epoch, whereas the GPT2-RMBA(0.05) model shows an improvement in accuracy after 0.25 epoch. The results indicate that the model utilizing RMBA achieves the highest data efficiency, reducing the number of iterations required to attain generalization by about 75% compared to the baseline model.

To verify whether these algorithms enable the model to learn the essence of the reasoning process, we inserted several random tokens at random positions in each test sentence and recorded the accuracy of the different models. Fig. 8(c) shows the accuracy of each model outputting the correct

answer as the number of random tokens increases. The IMBA models exhibit the least stability, while the RMBA models maintain the same level of stability as the baseline.

(a)

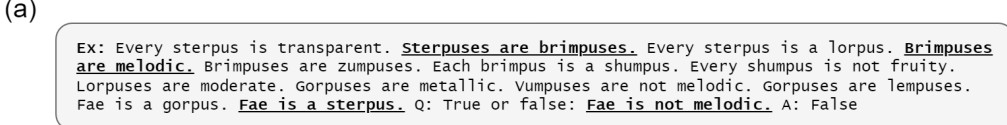

(b)                                                        (c)

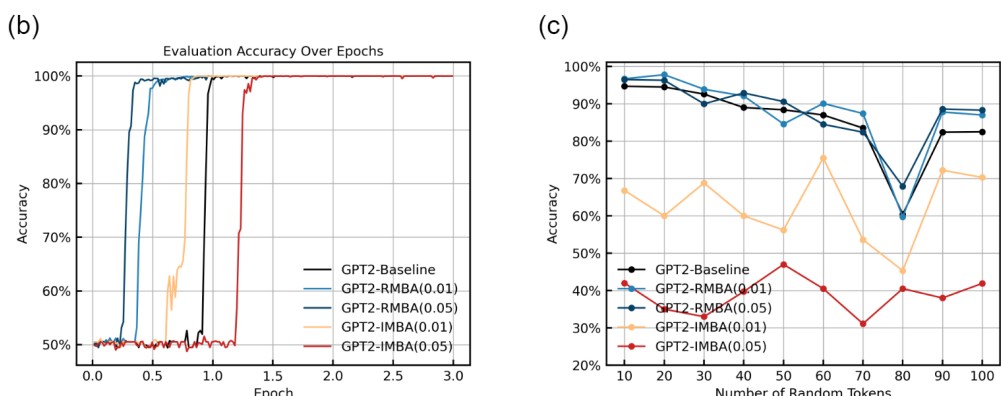

Figure 8: (a) Example of PrOntoQA dataset. (b) Accuracy curve of GPT-2 trained with RMBA and IMBA. For each algorithm, we initialize the learnable parameter $\alpha^{(l)}, \beta^{(l)}$ at 0.01 and 0.05. The GPT2-baseline model begins to generalize after approximately 1 epoch, whereas the GPT2-RMBA(0.05) model shows an improvement in accuracy after 0.25 epoch. Loss curve can be found in Appendix F. (c) Stability of different GPT-2 models. We inserted several random tokens at random positions in each test sentence and recorded the accuracy of the different models.

## 6 DISCUSSION

**Conclusion.** In this study, we investigated the buffer mechanism employed by Transformer models when performing symbolic multi-step reasoning using the vertical thinking strategy and horizontal thinking strategy. When utilizing the vertical thinking strategy, the model stores different intermediate results in separate buffers and transfers the reasoning results with same-token matching. In contrast, when applying the horizontal thinking strategy, the model reuses the existing buffers to store intermediate results, enabling reasoning over an arbitrary number of steps. We validated that the buffer mechanism is a key factor in enabling the model's ID and OOD generalization capabilities. Based on the buffer mechanism, we proposed a tailored approach, RMBA, to enhance the model's multi-step reasoning ability, significantly improving data efficiency when training GPT-2 on PrOntoQA datasets.

**Limitations and Future Work.** Our current work lacks in-depth theoretical analysis. In future work, we aim to conduct deeper theoretical modeling for multi-step reasoning problems and extend the buffer mechanism to other types of multi-step reasoning tasks.

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

# A  RELATED WORK

**Language model reasoning** There have been numerous experimental and theoretical studies on language model reasoning. Abbe et al. (2022) examines the reasoning capabilities of neural networks using the Pointer Value Retrieval (PVR) benchmark, which was originally introduced in Zhang et al. (2021). Sharma et al. (2023) demonstrates that applying low-rank approximation of the certain weight layers to themselves can enhance reasoning performance across various tasks, and Chen et al. (2024a) explains this phenomenon in a two-layer Transformers model. Wang et al. (2024a) investigate both ID and OOD reasoning abilities on two synthetic tasks involving composition and comparison. Jiang et al. (2024) reveals that the reasoning process in Large Language Models (LLMs) is influenced by token bias and that these models continue to face challenges when dealing with more complex logical tasks. Abbe et al. (2024) introduces the distribution locality and shows Transformers require low locality. Zhang et al. (2024b); Luo et al. (2021); Zhou et al. (2022) shows that small initialization can facilitate model reasoning. Brinkmann et al. (2024) investigates the mechanism by which language models output intermediate reasoning paths in multi-step reasoning tasks. Aubry et al. (2024) uncover a transformer block coupling phenomenon in a variety of LLMs by tracing the trajectories of individual tokens as they pass through transformer blocks.

Recently, the multi-hop reasoning abilities of models have garnered attention in the LLM field. Kil et al. (2024); Li et al. (2024a) promote the use of Chain-of-Thought (CoT) reasoning by models to perform multi-step thinking through appropriate prompt engineering. Zhang et al. (2023) propose a new retrieval framework for multiple reasoning paths. Dar et al. (2022); Li et al. (2024b); Yang et al. (2024); Shalev et al. (2024) pinpoint where models execute multi-step reasoning by causal intervention and observing neuron activation. Biran et al. (2024) introduce a method to enhance models' reasoning abilities by repeatedly invoking intermediate layers. These works, especially the causal intervention experiments, have inspired our research. However, existing studies primarily conduct macro-level statistical analyses on actual complex large language models. While causal intervention methods can help us identify critical paths, our work builds upon this foundation by further exploring how models leverage their own weights to generate these key paths. Experiments on our symbolic datasets also facilitate more in-depth experimental and theoretical investigations.

**Understanding the mechanism of neural model** Our work builds upon previous studies on the attention mechanism (Voita et al., 2019; Vig, 2019; Kovaleva et al., 2019; Kobayashi et al., 2020). Numerous researchers have proposed various approaches to identify the roles of different heads in Transformers (Vig et al., 2020; Jeoung & Diesner, 2022; Wang et al., 2022; Conmy et al., 2023; Merullo et al., 2023; Guo et al., 2023; Wang & Weinan, 2024; Amsel et al., 2024; Li et al., 2024c; Wang et al., 2024c). These methods predominantly employ the concept of perturbation. Similar to the observations made by Wang et al. (2023) and Dutta et al. (2024), who noted that large language models typically perform information aggregation in shallow layers and information induction in deeper layers, we have also observed comparable phenomena in our study. The idea of symbolic datasets is inspired by Poli et al. (2024); Zhang et al. (2024c). There have also been some insightful theoretical works on feedforward neural networks. A series of studies have explored neural network preferences and generalization from the perspectives of regularization and frequency, etc. (Xu et al., 2019; Wang et al., 2024b; Jacot et al., 2018; 2020; Arora et al., 2019a; 2018). And Wu & Su (2023); Wang & Wu; Arora et al. (2022); Li et al. (2021); Wu et al. (2018); Zhu et al. (2018); Arora et al. (2019b) investigates the dynamical behavior of neural networks, while Ren et al. (2024) examines the factors influencing neural network generalization.

**In-context learning and induction head** Our work primarily investigates the model's ability to perform in-context learning. The concept of in-context learning (ICL) was first introduced by Brown et al. (2020). Since then, a series of studies have utilized induction heads to investigate ICL, yielding remarkable research outcomes (Olsson et al., 2022; Garg et al., 2022; Wang et al., 2022; Müller et al., 2021; Goldowsky-Dill et al., 2023; Bietti et al., 2024; Nichani et al., 2024; Edelman et al., 2024; Chen et al., 2024b; Todd et al., 2023; Chen & Zou, 2024). It is worth noting that induction heads can be considered as a special case of multi-step reasoning tasks with reasoning step equals to 1. However, multi-step reasoning is not a simple linear combination of single-step reasoning. In our work, we study the mechanism that enables multi-step reasoning, which has not been explored in previous studies.

## B EXPERIMENTAL SETTINGS

In our experiments (Section 3), the vocabulary size is set to $d = 201$, and the hidden space dimension is set to $d_m = 400$ and $d_q = d_k = d_v = 64$. We use 200,000 2-step reasoning sequences (with sequence length equal to 13). The learning rate is set to 2e-5 and linearly warms up to 1e-4 within 400 epochs and then decays to 1e-5 within 3600 epochs. The batch size is set to 100. We use the AdamW optimizer with default parameters as set in PyTorch 2.3.0. We employ a gradient clipping strategy with `torch.nn.utils.clip_grad_norm_(model.parameters(), max_norm=1)`.

The experiments were conducted on a server with the following configuration:

- 64 AMD EPYC 7742 64-Core Processor.
- 256GB of total system memory.
- 2 NVIDIA A100 GPUs with 40GB of video memory each and 8 NVIDIA GeForce RTX 4080 GPUs with 16GB of video memory each.
- The experiments were run using the Ubuntu 22.04 LTS operating system.

The task shown above can be completed within 2 hours with a single NVIDIA A100 GPU. For other more complex examples, they can be finished within 24 hours.

## C PROOF OF LEMMA 1

**Lemma 1** *Suppose token $\boldsymbol{x} = \sum_{i=1}^{n} \boldsymbol{a}_i \boldsymbol{W}_i \in \mathbb{R}^{d_m}$ and token $\boldsymbol{y} = \sum_{i=1}^{n} \boldsymbol{b}_i \boldsymbol{W}_i \in \mathbb{R}^{d_m}$, where $\boldsymbol{a}_i, \boldsymbol{b}_i \in \mathbb{R}^{d_m}$, $\boldsymbol{W}_i \in \mathbb{R}^{d_m \times d_m}$, $i = 1, 2, \cdots, n$. Each element of $\{\boldsymbol{a}_i\}_{i=1}^{n}$, $\{\boldsymbol{b}_i\}_{i=1}^{n}$ and $\{\boldsymbol{W}_i\}_{i=1}^{n}$ follows a normal distribution $\mathcal{N}(0, 1/d_m)$ and independent to others. Then, we have:*

$$\boldsymbol{x} \boldsymbol{W}_i^\mathsf{T} = \boldsymbol{a}_i + \mathcal{O}\left(\sqrt{\frac{n}{d_m}}\right), \quad \boldsymbol{y} \boldsymbol{W}_j^\mathsf{T} = \boldsymbol{b}_j + \mathcal{O}\left(\sqrt{\frac{n}{d_m}}\right), \tag{16}$$

$$\boldsymbol{x} \boldsymbol{W}_i^\mathsf{T} \boldsymbol{W}_j \boldsymbol{y}^\mathsf{T} = \boldsymbol{a}_i \boldsymbol{b}_j^\mathsf{T} + \mathcal{O}\left(\frac{n}{\sqrt{d_m}}\right). \tag{17}$$

*Proof.* We first show that $\boldsymbol{W}_i \boldsymbol{W}_j^\mathsf{T}$ (denoted as $\boldsymbol{Z}^{(i,j)}$) is also a random matrix with elements following a normal distribution $\mathcal{N}(0, 1/d_m)$ when $i \neq j$. In fact,

$$\mathbb{E}[(\boldsymbol{W}_i \boldsymbol{W}_j^\mathsf{T})_{s,t}] = \mathbb{E}[\sum_{k=1}^{d_m} (\boldsymbol{W}_i)_{sk} (\boldsymbol{W}_j)_{kt}] = \sum_{k=1}^{d_m} \mathbb{E}[(\boldsymbol{W}_i)_{sk}] \mathbb{E}[(\boldsymbol{W}_j)_{kt}] = 0,$$

$$\mathrm{Var}\left[(\boldsymbol{W}_i \boldsymbol{W}_j^\mathsf{T})_{s,t}\right] = \mathbb{E}\left[\left(\sum_{k=1}^{d_m} (\boldsymbol{W}_i)_{sk} (\boldsymbol{W}_j)_{kt}\right)^2\right]$$

$$= \sum_{k=1}^{d_m} \mathbb{E}\left[(\boldsymbol{W}_i)_{sk}^2\right] \mathbb{E}\left[(\boldsymbol{W}_j)_{kt}^2\right] = d_m \times (\frac{1}{d_m})^2 = \frac{1}{d_m},$$

which indicate $\{\boldsymbol{Z}^{(i,j)}\}$ follows the same distribution as $\{\boldsymbol{W}_i\}_{i=1}^{n}$. Therefore,

$$\mathbb{E}_{\{\boldsymbol{W}_j\}}\left[(\boldsymbol{x} \boldsymbol{W}_i^\mathsf{T})_t\right] = \sum_{j=1}^{n} \mathbb{E}_{\{\boldsymbol{Z}^{(j,i)}\}_j}\left[(\boldsymbol{a}_j \boldsymbol{Z}^{(j,i)})_t\right]$$

$$= \sum_{k=1}^{d_m} (\boldsymbol{a}_i)_k \mathbb{E}\left[(\boldsymbol{Z}^{(i,i)})_{kt}\right] + \sum_{\substack{j=1 \\ j \neq i}}^{n} \sum_{k=1}^{d_m} (\boldsymbol{a}_j)_k \mathbb{E}\left[(\boldsymbol{Z}^{(j,i)})_{kt}\right]$$

$$= \sum_{k=1}^{d_m} (\boldsymbol{a}_i)_k \mathbb{E}\left[(\boldsymbol{Z}^{(i,i)})_{kt}\right] = (\boldsymbol{a}_i)_t,$$

$$\text{Var}_{\{\boldsymbol{W}_j\}}\left[(\boldsymbol{x}\boldsymbol{W}_i^\mathsf{T})_t\right] = \sum_{j=1}^{n} \text{Var}_{\{\boldsymbol{Z}^{(j,i)}\}_j}\left[(\boldsymbol{a}_j\boldsymbol{Z}^{(j,i)})_t\right]$$

$$= \sum_{k=1}^{d_m}(\boldsymbol{a}_i)_k^2\text{Var}\left[(\boldsymbol{Z}^{(i,i)})_{kt}\right] + \sum_{\substack{j=1\\j\neq i}}^{n}\sum_{k=1}^{d_m}(\boldsymbol{a}_j)_k^2\text{Var}\left[(\boldsymbol{Z}^{(j,i)})_{kt}\right]$$

$$= \sum_{k=1}^{d_m}(\boldsymbol{a}_i)_k^2\text{Var}\left[(\boldsymbol{Z}^{(i,i)})_{kt}\right] + \frac{1}{d_m}\sum_{\substack{j=1\\j\neq i}}^{n}\sum_{k=1}^{d_m}(\boldsymbol{a}_j)_k^2$$

$$= (\boldsymbol{a}_i)_t^2\text{Var}\left[(\boldsymbol{Z}^{(i,i)})_{tt}\right] + \sum_{\substack{k=1\\k\neq t}}^{d_m}(\boldsymbol{a}_i)_k^2\text{Var}\left[(\boldsymbol{Z}^{(i,i)})_{kt}\right] + \frac{1}{d_m}\sum_{\substack{j=1\\j\neq i}}^{n}\sum_{k=1}^{d_m}(\boldsymbol{a}_j)_k^2$$

$$= \frac{2}{d_m}(\boldsymbol{a}_i)_t^2 + \frac{1}{d_m}\sum_{\substack{k=1\\k\neq t}}^{d_m}(\boldsymbol{a}_i)_k^2 + \frac{1}{d_m}\sum_{\substack{j=1\\j\neq i}}^{n}\sum_{k=1}^{d_m}(\boldsymbol{a}_j)_k^2$$

$$= \frac{1}{d_m}(\boldsymbol{a}_i)_t^2 + \frac{1}{d_m}\sum_{j=1}^{n}\sum_{k=1}^{d_m}(\boldsymbol{a}_j)_k^2.$$

Therefore, $\text{Var}\left[(\boldsymbol{x}\boldsymbol{W}_i^\mathsf{T})_t\right] = \text{Var}_{\boldsymbol{a}}\left[\text{Var}_{\{\boldsymbol{W}_j\}}\left[(\boldsymbol{x}\boldsymbol{W}_i^\mathsf{T})_t\right]\right] = n/d_m + 1/d_m^2$. And Chebyshev's inequality implies that $\boldsymbol{x}\boldsymbol{W}_i^\mathsf{T} = \boldsymbol{a}_i + \mathcal{O}\left(\sqrt{\frac{n}{d_m}}\right)$, which also holds for $\boldsymbol{y}\boldsymbol{W}_j^\mathsf{T}$.

Assume that $\boldsymbol{x}\boldsymbol{W}_i^\mathsf{T} = \boldsymbol{a}_i + \boldsymbol{z}_1$, $\boldsymbol{y}\boldsymbol{W}_j^\mathsf{T} = \boldsymbol{b}_j + \boldsymbol{z}_2$, $\boldsymbol{z}_1, \boldsymbol{z}_2$ are random vector with the elements follow the normal distribution $\mathcal{N}(n, 1/d_m)$, then,

$$\text{Var}\left[\boldsymbol{x}\boldsymbol{W}_i^\mathsf{T}\boldsymbol{W}_j\boldsymbol{y}^\mathsf{T}\right] = \text{Var}\left[\boldsymbol{a}_i\boldsymbol{b}_j^\mathsf{T} + \boldsymbol{a}_i\boldsymbol{z}_2^\mathsf{T} + \boldsymbol{z}_1\boldsymbol{b}_j^\mathsf{T} + \boldsymbol{z}_1\boldsymbol{z}_2^\mathsf{T}\right]$$

$$= d_m \times \frac{1}{d_m} \times \frac{1}{d_m} + 2 \times d_m \times \frac{1}{d_m} \times \frac{n}{d_m} + d_m \times \frac{n}{d_m} \times \frac{n}{d_m} = \frac{(n+1)}{d_m}.$$

Thus we have $\boldsymbol{x}\boldsymbol{W}_i^\mathsf{T}\boldsymbol{W}_j\boldsymbol{y}^\mathsf{T} = \boldsymbol{a}_i\boldsymbol{b}_j^\mathsf{T} + \mathcal{O}\left(\frac{n}{\sqrt{d_m}}\right)$. $\qquad\square$

# D    FURTHER DISCUSSION ON THE VERTICAL THINKING STRATEGY

## D.1    INFORMATION FUSION

In our symbolic dataset task, as mentioned in Section. 3, the first layer facilitates the fusion of token information at odd and even positions. We find that positional encoding plays a crucial role in the features of the first layer of attention. Fig. 9(a)(b) illustrates a comparison between the original attention mechanism and the positional attention mechanism calculated with eq. 18. As shown, there is minimal difference between the two approaches.

$$\mathcal{A}^{(0)}(\boldsymbol{X}^{\text{pos}}) = \text{softmax}\left(\frac{\text{mask}(\boldsymbol{X}^{\text{pos}}\boldsymbol{W}^{qk}\boldsymbol{X}^{\text{pos},\mathsf{T}})}{\sqrt{d_k}}\right). \tag{18}$$

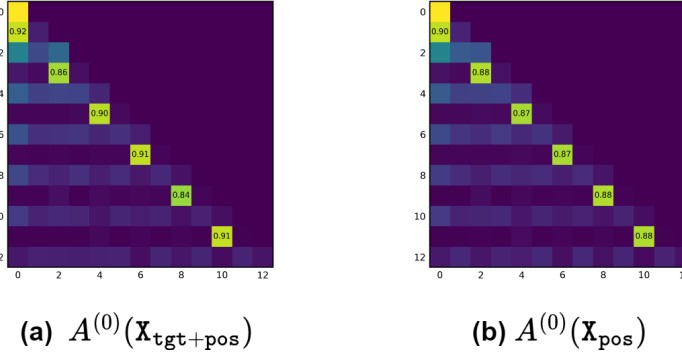

**(a)** $A^{(0)}(\mathbf{X_{tgt+pos}})$          **(b)** $A^{(0)}(\mathbf{X_{pos}})$

Figure 9: A comparison between the original attention (a) and the positional attention (b) of the first Transformer block, where $\boldsymbol{X}_{tgt+pos} = \boldsymbol{X}_{tgt} + \boldsymbol{X}_{pos}$.

## D.2    DETAILED MATCHING MATRIX

In Section. 3, for simplicity of analysis, we ignored the impact of the feedforward layer. Here, we define a detailed version of the matching matrix as follows:

$$\tilde{h}^{(1)}(\boldsymbol{X}) = f^{(0)}(\boldsymbol{X})\boldsymbol{W}^{q(1),\mathsf{T}}[f^{(0)}(\boldsymbol{X}\boldsymbol{W}^{vo(0)})\boldsymbol{W}^{k(1),\mathsf{T}}]^{\mathsf{T}} \tag{19}$$

$$\tilde{h}^{(2)}(\boldsymbol{X}) = f^{(1)}[f^{(0)}(\boldsymbol{X})\boldsymbol{W}^{vo(1)}]\boldsymbol{W}^{q(2),\mathsf{T}}\left[f^{(1)} \circ f^{(0)}(\boldsymbol{X}\boldsymbol{W}^{vo(0)})\boldsymbol{W}^{k(2),\mathsf{T}}\right]^{\mathsf{T}}. \tag{20}$$

As shown in Fig. 10, the detailed matching matrices still maintain the diagonal element property in most cases, even for the out-of-distribution tokens.

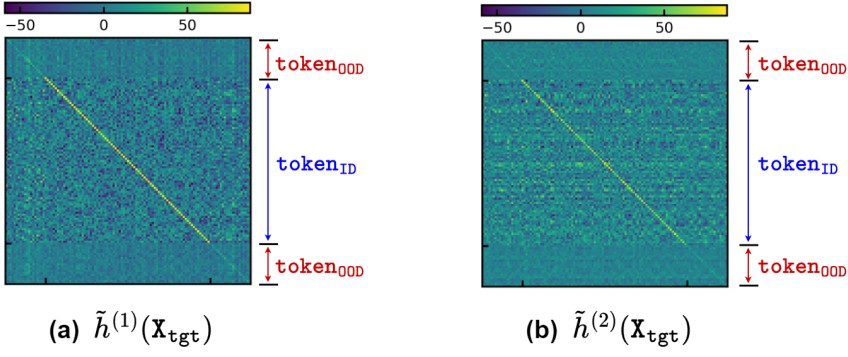

**(a)** $\tilde{h}^{(1)}(\mathbf{X_{tgt}})$          **(b)** $\tilde{h}^{(2)}(\mathbf{X_{tgt}})$

Figure 10: (a) Heatmap of $\tilde{h}^{(1)}(\boldsymbol{X}_{tgt})$ and $\tilde{h}^{(2)}(\boldsymbol{X}_{tgt})$. The diagonal elements exhibit the largest values, confirming the matching operation.

### D.3 INDEPENDENCE OF BUFFERS

To verify the independence between buffers, we computed and visualized the cosine similarity between row vectors of different buffers ($\boldsymbol{W}^{vo(0)}, \boldsymbol{W}^{vo(1)}, \boldsymbol{W}^{vo(2)}$ and $\boldsymbol{I}$). As shown in Fig. 11, apart from exhibiting a certain similarity within itself (so called condense phenomenon(Luo et al., 2021)), each buffer remains nearly orthogonal to the others.

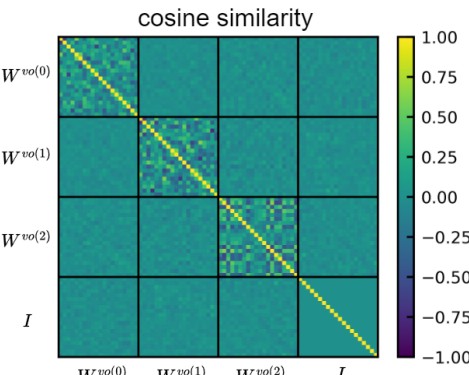

Figure 11: The cosine similarity between row vectors of different buffers ($\boldsymbol{W}^{vo(0)}, \boldsymbol{W}^{vo(1)}, \boldsymbol{W}^{vo(2)}$ and $\boldsymbol{I}$).

### D.4 WEIGHT CONSTRUCTION METHOD FOR MULTI-STEP REASONING NETWORKS

In Section. 3, we mentioned that by setting the weights in the following manner, we can enable an $L$-layer Transformer model to possess $(L-1)$-step reasoning capability.

$$\boldsymbol{W}^{q(0)} = \boldsymbol{W}^{q(1)} = I, \ \boldsymbol{W}^{q(l)} = \boldsymbol{W}^{vo(l-1),\mathsf{T}}, \ l \geq 2, \tag{21}$$

$$\boldsymbol{W}^{k(0)} = \sum_{i=1}^{[l_{\text{seq}}/2]} \boldsymbol{p}_{2i}\boldsymbol{p}_{2i-1}^{\mathsf{T}}, \ \boldsymbol{W}^{k(l)} = \boldsymbol{W}^{vo(0),\mathsf{T}}, \ l \geq 1, \tag{22}$$

where $\{\boldsymbol{W}^{vo(l)}\}_{l=1}^{L}$ are random matrices and the projection weight $\boldsymbol{W}^{p} = \boldsymbol{W}^{vo(L),\mathsf{T}}\boldsymbol{W}^{\text{emb},\mathsf{T}}$.

The specific construction method is as follows: To ensure that each buffer in the model has adequate robustness against interference, we set $d_m = d_q = d_k = d_v = 10000$. The feedforward layers are assigned zero weights so that the residual connection dominates. Since all the weight matrices we use are untrained random matrices, the layer normalization will have no effect. Fig. 12 shows the multi-step reasoning ability of an 8-layer Transformer. We tested natural order, reverse order, random order sentences, and sentences with inserted irrelevant tokens (i.e., token [20]), and the model was able to output the correct answer [8] in all cases. Each sentence begins with token [20] to prevent $\mathcal{A}_{0,0}^{(l)}$ always equals to 1, which could affect the buffer.

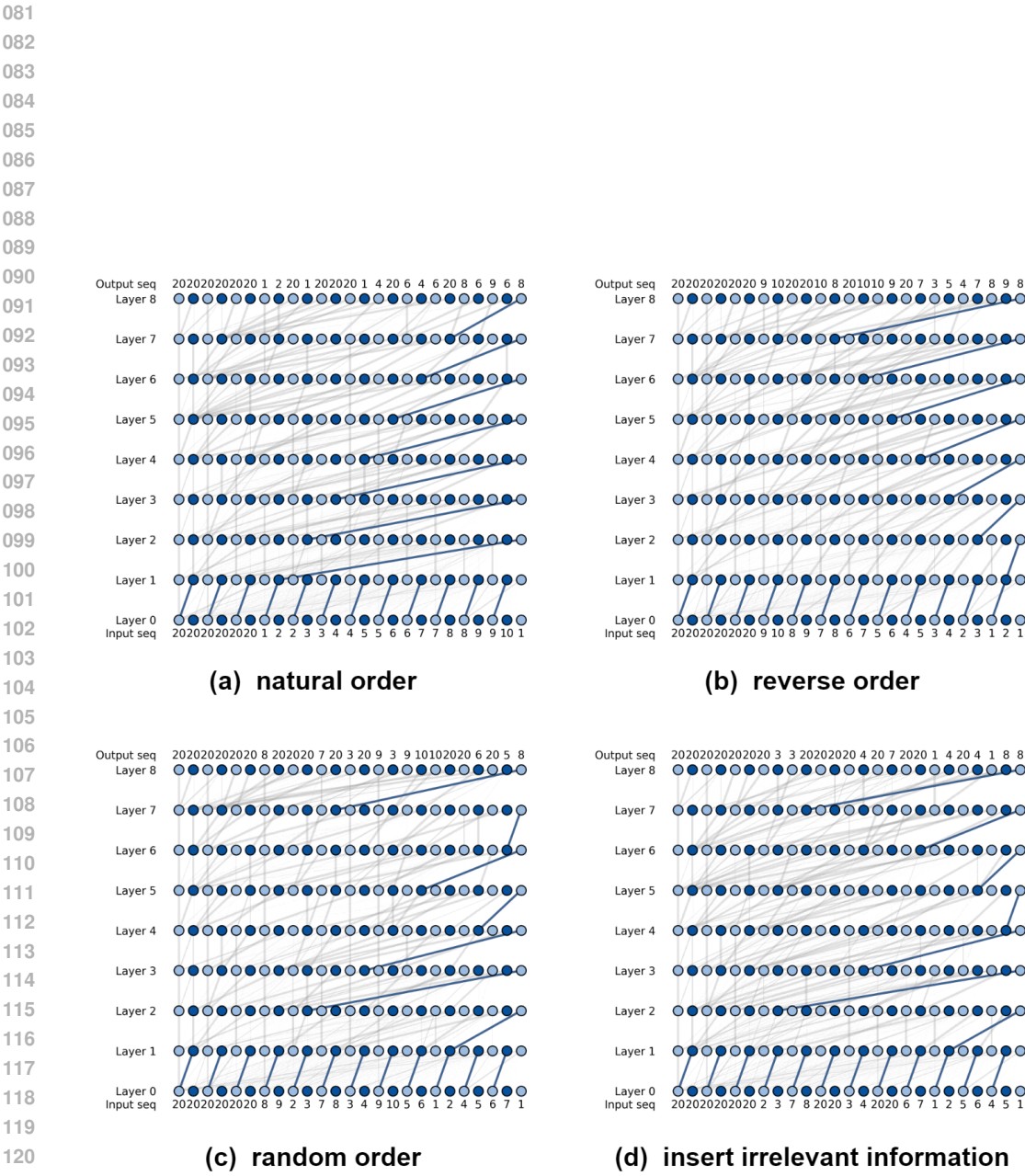

Figure 12: The test results for the 8-layer Transformer we constructed. The gray lines represent attention values that do not affect the final outcome. The width of all lines is positively correlated with the attention weights.

# E PARALLEL REASONING AND ITS POSSIBLE MECHANISM

In this section, we investigate a natural question arising from Section 3: Can the model still perform reasoning through the vertical thinking strategy when the number of reasoning steps required by the task exceeds the number of reasoning layers? Our experimental studies reveal that, in such cases, the model exhibits a parallel reasoning strategy. Based on these observations, we hypothesize that the upper bound of the model's reasoning capability will grow exponentially with the depth of the model. We have conducted preliminary verification of this hypothesis using a simple program.

**Possible Mechanism of Parallel Reasoning.** Fig. 13(a) illustrates the information flow of a 4-layer model completing 4-step reasoning. We observe that when the reasoning steps exceed or equal the number of model layers, the model performs reasoning parallelly in one layer. In the 2nd layer, different information undergoes two reasoning operations, enabling the subsequent three layers to achieve 4-step reasoning. Moreover, starting from the 2nd layer, the model's reasoning approach transits from matching tokens' values to matching tokens' positions. Token value matching is only utilized in the 1st layer.

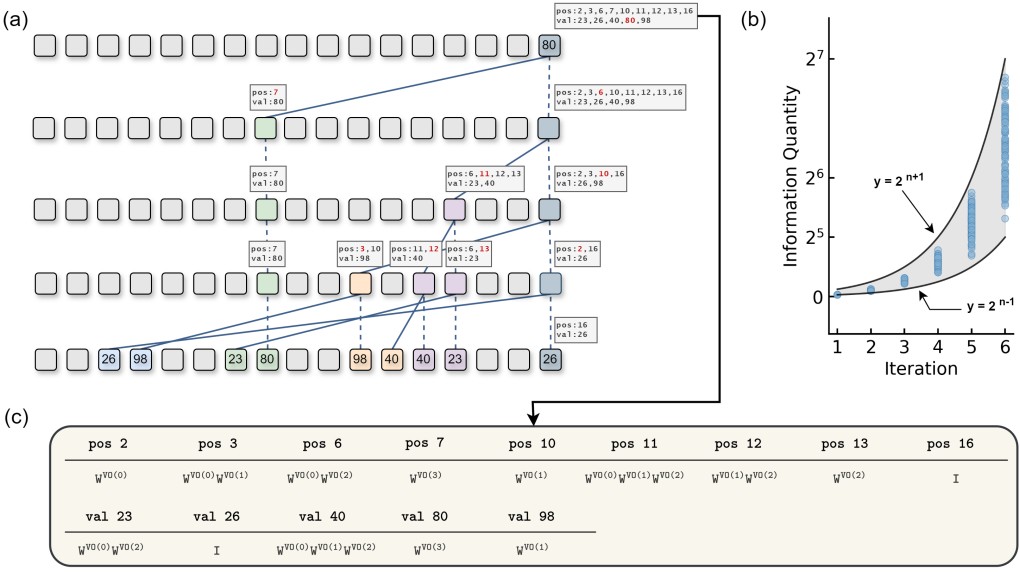

Figure 13: (a) Information flow of a 4-layer Transformer model completing 4-step reasoning. (b) Number of stored information tokens in the last position with respect to the number of iterations when propagating information based on the Info-Prop Rules. We randomly selected 1000 sentences for the simulation. The blue dots represent the simulation results. (c) The coefficients by which each information is multiplied.

Based on the understanding of information propagation, at the $l$-th layer, if the information is transmitted through attention, it should be multiplied by a coefficient $W^{vo(l)}$, whereas no such term is required if it is transmitted through residual connections. Consequently, in Fig. 13(c), we present the coefficients by which each information is multiplied. It is observed that for each positional information, the coefficients are distinct, and the same holds for the value information. The variation in coefficients also indicates that the information is stored in different buffers, thereby possibly enabling parallel reasoning.

**Upper Bound of Model's Reasoning Ability.** Through the study of the buffer mechanism in Transformer models, we can estimate the upper bound of the reasoning ability for a general L-layer Transformer network. First, we assume that the model satisfies the following assumption.

**Assumption 1.** *Assume that the hidden space dimension $d_m$ is sufficiently large, allowing different information to be stored in independent buffers with little interference. Moreover, information transmission is achieved solely through the same-token matching of the Attention module, without considering the influence of the feedforward module.*

We regard $n$ tokens of a sequence as $n$ nodes, each storing position and value information. The Transformer model follows the **Information Propagation (Info-Prop) Rules** (denoting the node transmitting information as $i$ and the node receiving information as $j$, considering the existence of the attention mask, we require $i < j$):

- Rule 1: Odd positions can pass information to subsequent even positions, i.e., node $i$ stores an odd number $a$ in its position information, and node $j$ stores $a + 1$ in its position information.
- Rule 2: The position information stored in node $i$ and node $j$ has common elements.
- Rule 3: The value information stored in node $i$ and node $j$ has common elements.

If any of the above three rules are satisfied, we consider that the information from node $i$ can be transmitted to node $j$. When information transmission occurs, the position and value information stored in node $j$ will take the union of the corresponding information from node $i$ and node $j$.

Pseudocode 1 provides a detailed process of the Info-Prop Rules. Each Node[i] is a dictionary containing two attributes: pos (position) and val (value). Node[i][pos] and Node[i][val] are two sets. Initially, each set contains only its own information and will be updated according to the Info-Prop Rules. Fig. 13(b) shows the number of value information stored in the last token with respect to the number of iterations under our set of rules. It can be observed that the stored information exhibits an approximately exponential growth pattern with the number of iterations.

---

**Algorithm 1** Information Propagation Rules

---

1: Initialize an empty dictionary Node
2: **for** each token i in the sequence **do**
3:     Node[i][pos] ← i
4:     Node[i][val] ← token i
5: **end for**
6: Node_new ← Node
7: Information propagation
8: **for** each node i in Node **do**
9:     **for** each node j > i in Node **do**
10:         rule1 ← any even number $a \in$ Node[i][pos] satisfies $a + 1 \in$ Node[j][pos]
11:         rule2 ← Node[i][val] ∩ Node[j][val] $\neq \varnothing$
12:         rule3 ← Node[i][pos] ∩ Node[j][pos] $\neq \varnothing$
13:         **if** rule1 or rule2 or rule3 **then**
14:             Node_new[j][pos] ← Node_new[j][pos] ∪ Node[i][pos]
15:             Node_new[j][val] ← Node_new[j][val] ∪ Node[i][val]
16:         **end if**
17:     **end for**
18: **end for**

---

A possible explanation for this experimental result is that in the $L$-th layer, the buffers of the Transformer can be represented as $\{\prod_{l \in J} \boldsymbol{W}^{vo(l)} | J \subset \{0, 1, \cdots, L - 1\}\}$. Therefore, the maximum amount of information (target information and position information) in the last layer is $2^{L+1}$. In practice, the hidden space dimensions $d_m$, $d_q$, $d_k$, and $d_v$ of large language models are far from meeting the requirements of Assumption 1. Consequently, the actual reasoning ability of large language models is constrained by the hidden space dimensions.

## F    DETAILS FOR RMBA EXPERIMENT

This section provides supplementary details on the experimental setup for the RMBA experiment. We use a 3-layer single-head Transformer with $d_q = d_k = d_v = 64$ and $d_m = 400$. The training set consists of 30,000 2-step reasoning chains. We trained the model for 200 epochs in total. In this setting, the Transformers have poor generation ability even in the in-distribution test dataset.

We replace $\boldsymbol{W}^{qk(l)}$ and $\boldsymbol{W}^{vo(l)}$ with $\boldsymbol{W}^{qk(l)} + \alpha^{(l)} \boldsymbol{Z}^{(l-1)}$ and $\boldsymbol{W}^{vo(l)} + \beta^{(l)} \boldsymbol{Z}^{(l)}$, respectively, where $\alpha^{(l)}$ and $\beta^{(l)}$ are learnable parameters, and $\{\boldsymbol{Z}^{(l)}\}_{l=1}^{L}$ is a set of random matrix following $N(0, 1/d_m)$. Therefore, 6 extra learnable parameters are added to this 3-layer single-head model in total. Fig. 14 and Fig. 15 show the loss and accuracy of the Transformer under different settings. Fig. 16 shows the changes of the learnable parameters $\alpha^{(l)}$ and $\beta^{(l)}$ during training.

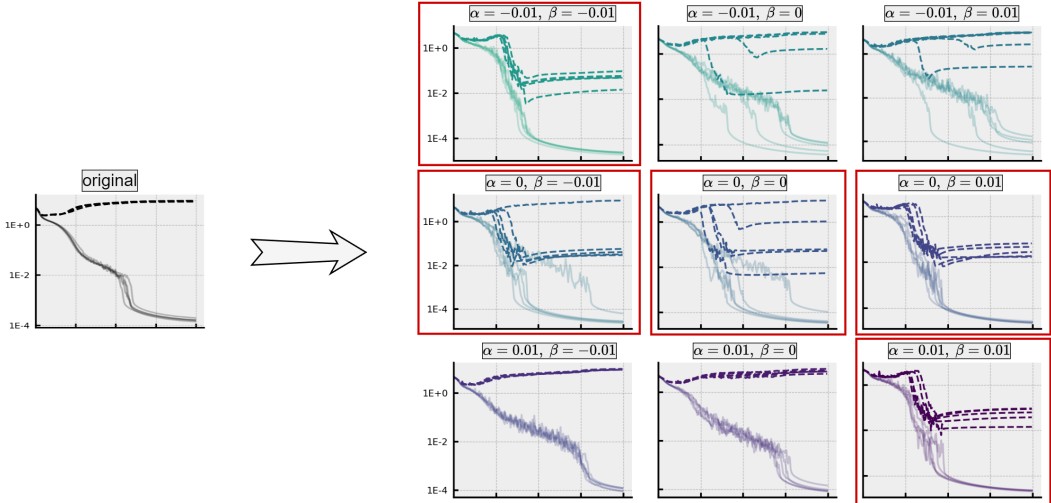

Figure 14: The impact of different learnable parameters' initial values, $\alpha^{(l)}$ and $\beta^{(l)}$, on the model's reasoning ability. The solid lines represent the training loss, while the dashed lines denote the test loss. Each experiment was conducted with 5 random seeds.

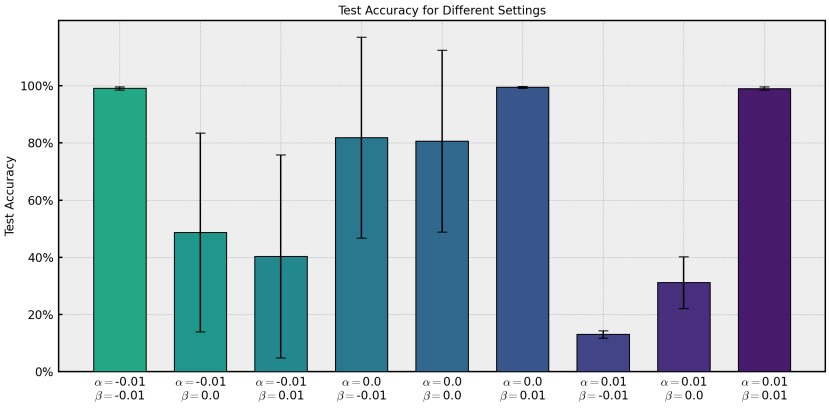

Figure 15: The average accuracy of different initial value of $\alpha^{(l)}$ and $\beta^{(l)}$.

We further tested the performance of the three algorithms under different hyperparameter configurations. We investigated the effects of weight decay, learning rate, and hidden dimension on training. For the RMBA and IMBA algorithms, we set $\alpha_{\text{ini}}^{(l)} = \beta_{\text{ini}}^{(l)} = 0.01$. As shown in Fig. 17, under various settings, the RMBA algorithm consistently facilitated the model's ability to generalize.

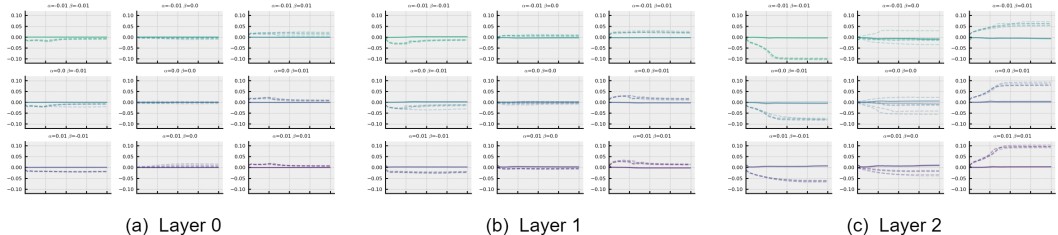

(a) Layer 0      (b) Layer 1      (c) Layer 2

Figure 16: Changes of the learnable parameters $\alpha^{(l)}$ and $\beta^{(l)}$ during training. The solid lines represent the $\alpha^{(l)}$, while the dashed lines denote the $\beta^{(l)}$. Each experiment was conducted with 5 random seeds.

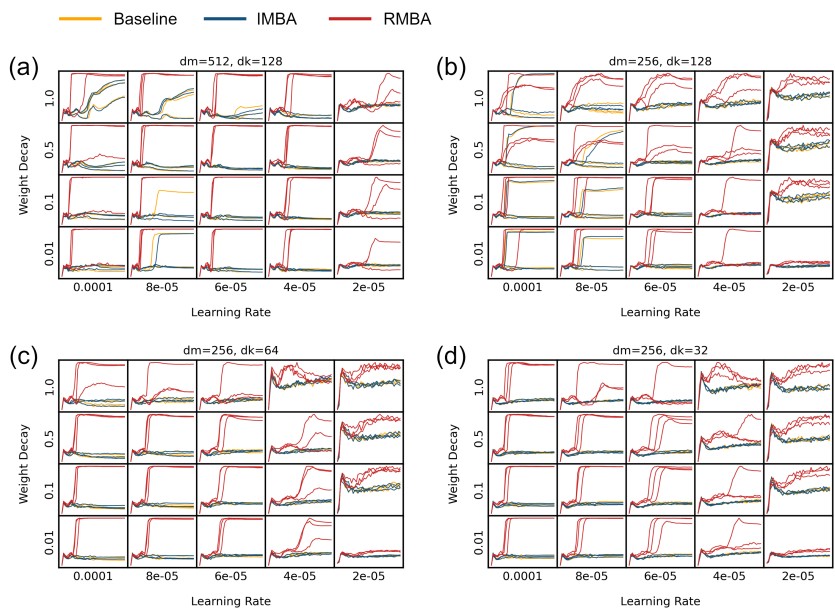

Figure 17: A comparison of the training results for RMBA, IMBA, and the Baseline model under different hyperparameters is presented. We investigated learning rates ranging from 2e-5 to 1e-4 and weight decay values from 0.01 to 1. We considered four configurations of the hidden space dimension. For each hyperparameter setting, we conducted experiments using 3 different random seeds, totaling 720 experiments.

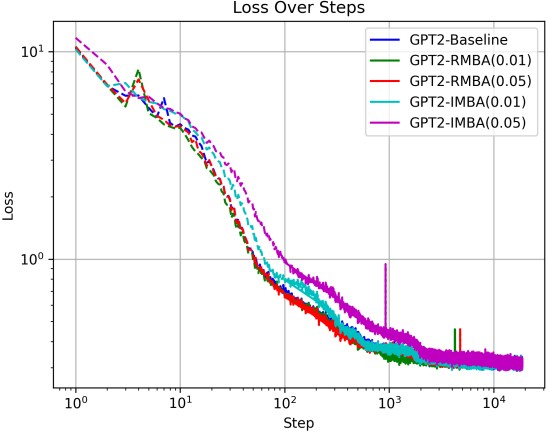

Figure 18: Loss curve of GPT-2 trained with RMBA and IMBA. (PrOntoQA dataset)

## G   DETAILS FOR HORIZONTAL THINKING STRATEGY

We trained a 2-layer Transformer model and investigated its ability to perform multi-step reasoning using lateral thinking. Specifically, we trained the Transformer model on 20,000 samples of length 13, each labeled with the result of a one-step reasoning process. The model parameters were set to $d_m = 400$ and $d_q = d_k = d_v = 256$, with a weight decay of 0.05. The matching matrix of the trained model is shown in Fig. 20(a).

As shown in Fig. 19, when we fed the model's output back into the model, it was able to generate the next step's reasoning result, even though it had never been exposed to sentences longer than length 13 during training. Fig. 20(b) shows the relationship between reasoning complexity (number of reasoning steps) and CoT accuracy. Our 2-layer model is able to maintain an accuracy of over 57.6% even when performing complex 6-step reasoning tasks.

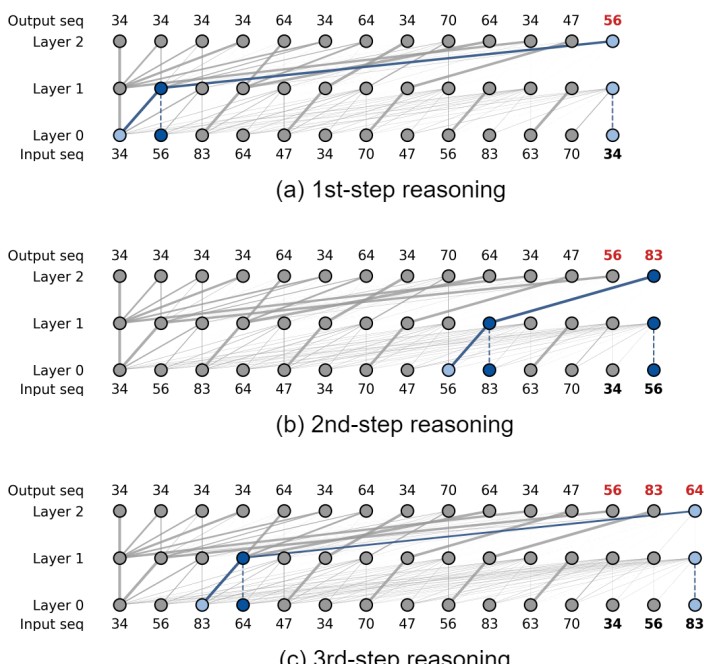

Figure 19: An illustration of the process of performing 3-step reasoning using a 2-layer Transformer model with Chain-of-Thought (CoT). The width of the connections in the diagram is based on the attention weights.

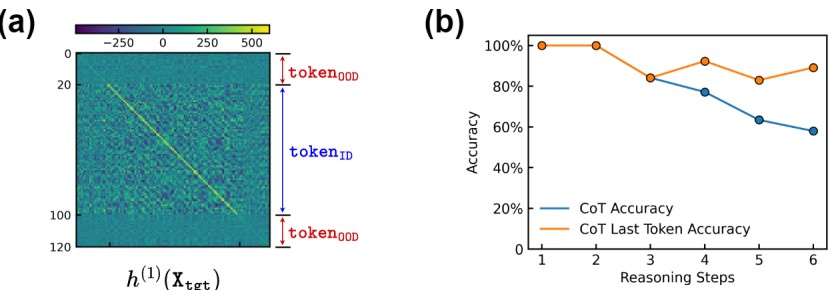

Figure 20: (a) Heatmap of matching matrix $h^{(1)}(\boldsymbol{X}_{tgt})$ for our CoT model. (b) The relationship between reasoning complexity (number of reasoning steps) and CoT accuracy. The CoT accuracy curve represents the performance when all reasoning steps are predicted correctly using CoT, while the CoT last token accuracy curve records the performance based solely on whether the final reasoning step is correct when the previous correct reasoning result is given.

# H MATCHING SCORE AND KERNEL SCORE FOR PHI-3 MODEL

In this section, we calculate the matching score and kernel score of the large language model Phi-3(Abdin et al., 2024).

We focus on whether $\boldsymbol{W}^{vo}$ and $\boldsymbol{W}^{qk}$ function as the information buffer and the information extractor, respectively. To simplify our analysis, we temporarily disregard the effects of the feed-forward layers. Following the method described in the main text, we compute $\text{Ker}^{(l_1,l_2)} = \boldsymbol{W}^{qk(l_1)}\boldsymbol{W}^{vo(l_2),\mathsf{T}}$ and observe whether it exhibits a dominant diagonal characteristic. For multi-head models, the above equation is modified as:

$$\text{Ker}^{(l_1,l_2)} = \left(\sum_h \boldsymbol{W}^{q(l_1,h)}\boldsymbol{W}^{k(l_1,h),\mathsf{T}}\right)\boldsymbol{W}^{vo(l_2),\mathsf{T}}. \tag{23}$$

We define the Kernel Score(KS) as

$$\text{KS}(\text{Ker}^{(l_1,l_2)}) = \text{Trace}(\sigma(\text{Ker}^{(l_1,l_2)}))/d_m, \tag{24}$$

which measures the ability of layer $l_1$ in the model to extract information cached at layer $l_2$. As shown in Fig. 21(a), when $l_1 \geq l_2$, the kernel score is nearly zero, which aligns with the logical sequence of information processing. When $l_1 < l_2$, the kernel score decreases as $(l_2 - l_1)$ increases, indicating that the model tends to extract the most recently acquired information for further processing. In Fig. 21(b), we plot $\sigma(\text{Ker}^{(l_1,l_2)})$ and highlight the regions where the Kernel Score $> 0.3$.

To further verify that the diagonal structure of $\text{Ker}^{(l_1,l_2)}$ arises from the alignment of model weights rather than the intrinsic diagonal structure of $\sum_h \boldsymbol{W}^{q(l_1,h)}\boldsymbol{W}^{k(l_1,h),\mathsf{T}}$ and $\boldsymbol{W}^{vo(l_2),\mathsf{T}}$, we conducted a small experiment, as illustrated in Fig. 21(d). In this experiment, we assume that both matrices $A$ and $B$ are noise-added identity matrices, where the noise scale is denoted by $\alpha$. We then compute the kernel scores of $A$, $B$, and their product $C = AB$. The results show that when $\max(\text{KS}(A),\text{KS}(B)) < 0.3$, the Kernel score of the product, $\text{KS}(C)$, is less than $0.025$. However, as shown in Fig. 21(c), we observe that for many heads, even when $\max(\text{KS}(\sum_h \boldsymbol{W}^{q(h)}\boldsymbol{W}^{k(h),\mathsf{T}}),\text{KS}(\boldsymbol{W}^{vo,\mathsf{T}})) < 0.3$, the $\text{Ker}^{(l_1,l_2)}$ still has a large kernel score. This indicates that the alignment of model weights is the key factor driving the diagonal structure of $\text{Ker}^{(l_1,l_2)}$. In Fig. 21(f), we present an example where both $\sum_h \boldsymbol{W}^{q(l_1,h)}\boldsymbol{W}^{k(l_1,h),\mathsf{T}}$ and $\boldsymbol{W}^{vo(l_2),\mathsf{T}}$ appear relatively disordered individually, but their product exhibits a clear diagonal structure.

Another straightforward method is to directly set the diagonal elements of $\sum_h \boldsymbol{W}^{q(l_1,h)}\boldsymbol{W}^{k(l_1,h),\mathsf{T}}$ and $\boldsymbol{W}^{vo(l_2),\mathsf{T}}$ to zero, and then calculate the kernel score based on the resulting $\tilde{\text{Ker}}^{(l_1,l_2)}$. Fig. 21(e) illustrates this result, showing that it is approximately the same as that in Fig. 21(a).

Moreover, without loss of generality, we consider the case that includes LayerNorm(LN) and feed-forward(FNN) layers. We compute the matching score for each head in each layer, with the specific calculation formula as follows:

$$\boldsymbol{X}^{vo} = \text{LN}_{\text{attn}}^{(l-1)}(\boldsymbol{X})\boldsymbol{W}^{v(l-1),\mathsf{T}}\boldsymbol{W}^{o(l-1),\mathsf{T}}, \tag{25}$$

$$\boldsymbol{X}^{vof} = \boldsymbol{X}^{vo} + \text{FNN}^{(l-1)}(\text{LN}_{\text{FNN}}^{(l-1)}(\boldsymbol{X}^{vo})), \tag{26}$$

$$\boldsymbol{X}^{vok(h)} = \text{LN}_{\text{attn}}^{(l)}(\boldsymbol{X}^{vof})\boldsymbol{W}^{k(l,h)}, \tag{27}$$

$$\boldsymbol{X}^f = \text{LN}_{\text{attn}}^{(l-1)}(\boldsymbol{X}) + \text{FNN}^{(l-1)}(\text{LN}_{\text{FNN}}^{(l-1)}(\text{LN}_{\text{attn}}^{(l-1)}(\boldsymbol{X}))), \tag{28}$$

$$\boldsymbol{X}^{q(h)} = \boldsymbol{X}^f \boldsymbol{W}^{q(l,h)}, \tag{29}$$

$$\text{matching matrix: } h^{(l,h)}(\boldsymbol{X}) = \boldsymbol{X}^{q(h)}\boldsymbol{X}^{vok(h),\mathsf{T}}. \tag{30}$$

We visualized the matching score of each head in each layer (Fig. 21(b)) and found that the matching scores were highest in layers 5 to 20. This aligns with the conclusion mentioned in Dutta et al. (2024), namely that the reasoning layers of large language models generally appear in the middle portion.

To further validate that Phi-3 might employ a buffer mechanism, we computed the pairwise cosine similarity of the matrices $\{\boldsymbol{W}^{vo(l)}\}$ (each matrix is flattened as a long vector). The results indicate that these matrices are nearly orthogonal to each other, suggesting that they can be treated as independent buffers.

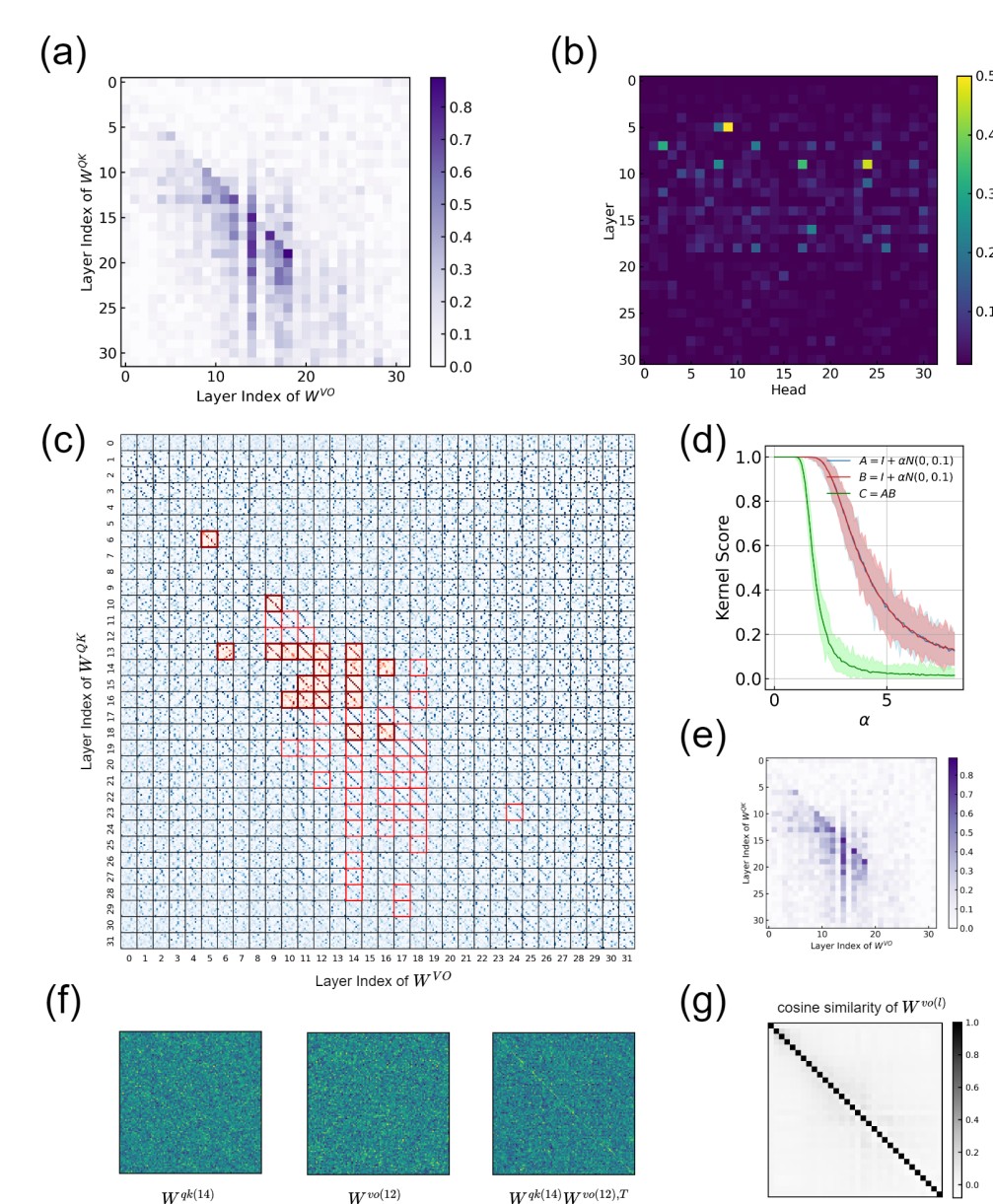

Figure 21: The calculation results of the kernel score and matching score for the Phi-3 model. (a) Visualization of $\sigma(\text{Ker}^{(l_1,l_2)})$. (b) Visualization of the matching score calculations for each head in each layer, indicating that reasoning is concentrated in the middle layers of the model. (c) Visualization of the kernel matrix between layers, where the subgraphs enclosed in red boxes correspond to $\text{KS}(\text{Ker}) > 0.3$ and the subgraphs enclosed in darkred boxes correspond to $\text{KS}(\text{Ker}) > 0.3$ but $\max(\text{KS}(\sum_h \boldsymbol{W}^{q(h)}\boldsymbol{W}^{k(h),\mathsf{T}}), \text{KS}(\boldsymbol{W}^{vo,\mathsf{T}})) < 0.3$. (d) The kernel score of a noise-added identity matrix(A and B) and the kernel score of the product of two noise-added identity matrices(C). It can be observed that for two unrelated matrices when their individual kernel scores are less than 0.3, the kernel score of their product is less than 0.025. (e) The kernel score obtained after setting the diagonal elements of $\sum_h \boldsymbol{W}^{q(h)}\boldsymbol{W}^{k(h),\mathsf{T}}$ and $\boldsymbol{W}^{vo,\mathsf{T}}$ to zero. (f) Visualization of the $\sum_h \boldsymbol{W}^{q(14,h)}\boldsymbol{W}^{k(14,h),\mathsf{T}}$ and the $\boldsymbol{W}^{vo(l_2),\mathsf{T}}$ in the Phi-3 model, along with their inner product. Despite their weak diagonal structure individually, their inner product exhibits a clear diagonal structure. (g) Cosine similarity of flattened $\boldsymbol{W}^{vo(l_1)}$ and $\boldsymbol{W}^{vo(l_2)}$, $l_1, l_2 \in \{0, \cdots, 31\}$.

# I  INTERACTION RESULTS WITH LARGE LANGUAGE MODELS

| 4-step reasoning (Correct: [r]) | | | | 3-step reasoning (Correct: [i]) | | | | 2-step reasoning (Correct: [e]) | | | |
|---|---|---|---|---|---|---|---|---|---|---|---|
| ChatGPT4 | ChatGPT4o | Claude3.5-haiku-20241022 | Clauld3.5-sonnet-20241022 | ChatGPT4 | ChatGPT4o | Claude3.5-haiku-20241022 | Clauld3.5-sonnet-20241022 | ChatGPT4 | ChatGPT4o | Claude3.5-haiku-20241022 | Clauld3.5-sonnet-20241022 |
| a | i | a | n | a | a | a | e | a | e | e | e |
| a | i | a | e | a | n | a | a | e | a | a | e |
| a | a | a | e | e | e | a | a | a | n | a | e |
| a | a | a | a | a | a | e | n | e | a | e | e |
| a | r | r | e | a | a | r | e | e | e | a | e |
| a | a | a | a | e | i | e | r | e | e | e | e |
| a | a | a | i | a | e | a | a | a | e | e | e |
| a | a | a | w | a | a | a | a | e | a | a | e |
| o | a | a | w | a | a | a | a | e | a | a | e |
| a | a | a | r | a | o | a | e | e | p | a | e |
| a | a | a | e | a | a | a | e | a | e | a | e |
| a | a | a | i | a | a | a | a | a | a | e | e |
| a | a | a | e | a | a | e | a | a | a | a | e |
| a | i | n | w | a | e | a | i | a | e | a | e |
| a | a | a | o | a | a | a | e | e | a | a | e |
| a | a | a | r | a | a | e | a | e | a | a | e |
| a | a | a | a | e | e | a | a | a | e | a | e |
| a | a | a | e | e | e | a | a | a | e | a | e |
| a | a | e | e | a | n | a | a | a | a | e | e |
| a | a | a | a | a | o | a | n | e | a | a | e |
| a | a | n | e | a | a | a | a | e | e | a | e |
| a | a | a | r | a | a | a | n | a | a | a | e |
| a | a | a | a | a | a | a | a | e | a | a | e |
| a | a | a | a | e | a | a | a | a | e | a | e |
| a | e | a | e | a | n | a | a | e | e | a | e |

Figure 22: Detailed interaction results with large models. For each type of reasoning task, we tested each large model 25 times. The versions of Claude are Claude 3.5-haiku-20241022 and Claude 3.5-sonnet-20241022.

## J CAUSAL INTERVENTION

This section presents causal intervention experiments conducted to verify that the model uses a buffer mechanism when performing symbolic multi-step reasoning tasks. We assume readers are familiar with the content of Section 3. The causal intervention experiments were conducted using a 3-layer Transformer model trained as described in Section 3.2.

First, we identified critical tokens and logical circuits by observing output changes when masking specific attention or residual paths. Fig. 23(left) illustrates the logical circuit of the 3-layer Transformer performing the symbolic 2-step reasoning task. Subsequently, we conducted causal intervention experiments on the information stored in each token.

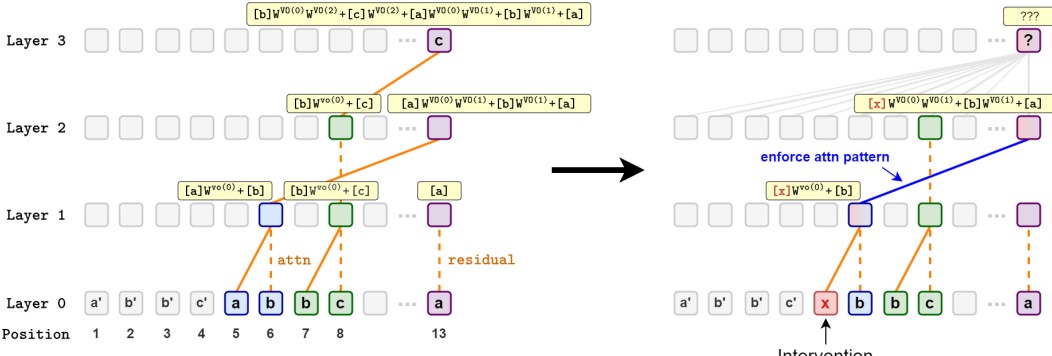

Figure 23: Logical circuit for 2-step reasoning (left) and illustration of causal intervention(right). We achieve the goal of intervening information stored in a specific buffer by modifying the input sequence and enforcing the same attention pattern as before. The figure illustrates how the information [a] stored in the final token's buffer $\boldsymbol{W}^{vo(0)}\boldsymbol{W}^{vo(1)}$ in layer 2 can be changed to [x].

Unlike prior works where causal intervention replaced all information within a token with alternative information(Feng & Steinhardt, 2023; Meng et al., 2022; Vig et al., 2020; Wang et al., 2024a), we refined the perturbation scope to target a specific buffer within a token. Specifically, we individually replaced the information in each buffer of critical tokens with alternative information to observe the model's output. For example, as shown in Fig. 23(right), suppose we want to change the information [a] stored in the final token's buffer $\boldsymbol{W}^{vo(0)}\boldsymbol{W}^{vo(1)}$ in layer 2 to [x]. This can be achieved by simply modifying the input sentence [a'][b'][b'][c'][a][b][b][c]...[a] to [a'][b'][b'][c'][x][b][b][c]...[a], and enforcing $\text{Attn}_{[13:]}^{(1)}$ same as before. Fig. 24 shows the intervention results for all buffers of all critical tokens.

The results reveal that, in layer $l$, the information stored in buffer $W^{vo(l-1)}$ of the final token is crucial. Modifying information in other buffers does not affect the model's output. Combined with the observation in Appendix D that the cosine similarities between $\boldsymbol{W}^{vo(0)}, \boldsymbol{W}^{vo(1)}, \boldsymbol{W}^{vo(2)}, \boldsymbol{I}$ are nearly zero, we can confidently assert that the model performs reasoning by leveraging different buffers. This experiment also rules out the possibility of the model only using an overwrite mechanism (as discussed in Section 4) to perform reasoning.

| Position | Buffer | Intervened Info | Method | Output |
|---|---|---|---|---|
| —— | —— | —— | Input [a'][b'][b'][c'][a][b][b][c]...[a] | [c] |
| Layer1 Token5 | $W^{vo(0)}$ | [a] → [x] | Input [a'][b'][b'][c'][x][b][b][c]...[a] | Random |
| Layer1 Token5 | I | [b] → [b'] | Input [a'][b'][b'][c'][a][b'][b][c]...[a] | [c'] |
| Layer1 Token8 | $W^{vo(0)}$ | [b] → [b'] | Input [a'][b'][b'][c'][a][b][b'][c]...[a] | Random |
| Layer1 Token8 | I | [c] → [x] | Input [a'][b'][b'][c'][a][b][b][x]...[a] | [x] |
| Layer2 Token8 | $W^{vo(0)}$ | [b] → [b'] | Input [a'][b'][b'][c'][a][b][b'][c]...[a] | Random |
| Layer2 Token8 | I | [c] → [x] | Input [a'][b'][b'][c'][a][b][b][x]...[a] | [x] |
| Layer2 Token13 | $W^{vo(0)}W^{vo(1)}$ | [a] → [x] | Input [a'][b'][b'][c'][x][b][b][c]...[a], enforce $\text{Attn}^{(1)}_{[13:]}$ same as before | [c] |
| Layer2 Token13 | $W^{vo(1)}$ | [b] → [b'] | Input [a'][b'][b'][c'][a][b'][b][c]...[a] | [c'] |
| Layer2 Token13 | $W^{vo(1)}$ | [b] → [x] | Input [a'][b'][b'][c'][a][x][b][c]...[a] | Random |
| Layer2 Token13 | I | [a] → [x] | Input [a'][b'][b'][c'][a][b][b][c]...[x], enforce $\text{Attn}^{(1)}_{[13:]}$ same as before | [c] |
| Layer3 Token13 | $W^{vo(0)}W^{vo(2)}$ | [b] → [x] | Input [a'][b'][b'][c'][a][b][x][c]...[a], enforce $\text{Attn}^{(2)}_{[13:]}$ same as before | [c] |
| Layer3 Token13 | $W^{vo(2)}$ | [c] → [x] | Input [a'][b'][b'][c'][a][b][b][x]...[a] | [x] |
| Layer3 Token13 | $W^{vo(0)}W^{vo(1)}$ | [a] → [x] | Input [a'][b'][b'][c'][x][b][b][c]...[a], enforce $\text{Attn}^{(1)}_{[13:]}$ same as before | [c] |
| Layer3 Token13 | $W^{vo(1)}$ | [b] → [x] | Input [a'][b'][b'][c'][a][x][b][c]...[a], enforce $\text{Attn}^{(2)}_{[13:]}$ same as before | [c] |
| Layer3 Token13 | I | [a] → [x] | Input [a'][b'][b'][c'][a][b][b][c]...[x], enforce $\text{Attn}^{(1)}_{[13:]}$ same as before | [c] |

Figure 24: Causal Intervention Experiment. We performed interventions on the information stored in every buffer of every critical token individually. Here, [x] represents a token that does not appear in the original input. For the final token, only modifying the buffer $W^{vo(l-1)}$ in layer $l$ affects the final output. In this experiment, the tokens in the original sentence are selected from the range [20, 40], while token [x] traverses the range [40, 100]. $\text{Attn}^{(l)}_{[13:]}$ refers to the attention score corresponding to the last token at layer $l$. For the original input, we have $\text{Attn}^{(1)}_{[13,6]} = 1$ and $\text{Attn}^{(2)}_{[13,8]} = 1$. Instances labeled as "Random" indicate that the output varies erratically as [x] changes. In all other cases, the probability of the model output deviating from the value presented in the table is less than 1e-15.

