# OpenReview forum: "The Buffer Mechanism for Multi-Step Information Reasoning in Language Models"
_ICLR.cc/2025/Conference — Submitted to ICLR 2025_

### Official Review · Reviewer_mhu8 · 2024-10-29

**Soundness:** 3
**Presentation:** 3
**Contribution:** 3
**Rating:** 6
**Confidence:** 3

**Summary:**

This paper analyzes the internal reasoning mechanisms of Transformers, from the perspective of vertical and horizontal thinking on a symbolic multi-step reasoning dataset. They analyze the concept of Buffer Mechanism, by which the model stores information and retrieves it through the query-key matrix. They explain how the model is leveraging this mechanism enhance the model’s multi-step reasoning capability. Finally, they show how their method reduces the cost of generalization.

**Strengths:**

1. Insightful Findings. The paper presents interesting findings on how the Transformers make use of the Buffer Mechanism.
2. Great Contribution. RMBA show that the authors can leverage the Buffer Mechanism to provide better efficiency.
3. Rigorous experimentation. The analysis and experiments show a careful detail and understanding of the processes.

**Weaknesses:**

1. Poor Presentation: The paper is hard to read in general. There are few explanation to the equations and Figures. The presentation of the work does not follow a good format. I believe the paper can have a much greater impact if the more attention is given to the paper writing and presentation.
2. Presentation of experiment results: The paper does clearly define the metrics to compare the results from IMBA and RMBA, which makes it hard to understand at first reading. For example, in Figure 7, they show that the IMBA model is unable to achieve a good test accuracy, but they do not quantify it or provide any further analysis.
3. Better Math Descriptions: Equations and Formulas are introduced without a former introduction and explanation.

**Questions:**

Some questions about the paper:
1. Can you explain why you claim 75% when compared to the baseline? I see the plots on Figure 8, but I cannot see the numbers to quantify this generalization myself.
2. Does the RMBA have some drawback during training? I assume the training process might be more unstable due the randomness of the matrix, is such behavior observed?

In general more clarity and better description of the experiments can help understanding the paper better. I miss a clear structure of the paper to understand it better the first time.

Some recommendations:
- Figure 2: It is missing an explanation of what (a) and (b) figures represent and what are the takeaways from those figures.
- Figure 5,6: Both are hard to read and not display the axes. I can deduct their meaning after reading the paper. But more clarity can help other readers.
- Figure 4: How does it provide a better understanding of the analysis?

---

> ### Author Response · Authors · 2024-11-18
>
> Thank you for your constructive comments on our paper. Below are our detailed responses to each of your points:
>
> **[R4W1] [Insufficient Explanations of Equations and Figures]**
>
> We have thoroughly revised the captions of every figure and table in the paper, adding detailed explanations to the flowcharts and supplementing the experimental figures with conclusions. Additionally, we have rewritten some paragraphs that might have been ambiguous to enhance readability. Due to the extensive nature of these changes, please refer to our revised manuscript for specific modifications.
>
> **[R4Q1] [75% Time Saving]**
>
> We apologize again for the lack of detailed explanation. As shown in Figure 8(b), after approximately one epoch, the accuracy of the baseline model rises above 95%, while the RMBA0.05 model reaches nearly the same accuracy after only about 0.25 epochs. Therefore, it saves about 75% of the training time. We have added the corresponding explanation in the new manuscript.
>
> **[R4Q2] [RMBA Stability]**
>
> According to our current experimental results, RMBA does not appear to have significant stability issues. Firstly, in Figure 8(c), we show the results after adding perturbations to different models; the experiments indicate that the RMBA algorithm has stability comparable to the baseline. We have also added the training loss curves of the PrOntoQA task in the appendix of the new manuscript, which also support our viewpoint.
>
> From a theoretical perspective, RMBA can be understood as a special form of residual connection (just multiplied by a constant matrix), and residual connections are known to promote stability. We speculate that this may be one reason why RMBA does not lead to decreased stability.
>
> **[R4Q3] [Recommendations]**
>
> We will comprehensively revise the figure captions according to your suggestions.
>
> **[Other Modifications]**
>
> In addition, we have incorporated suggestions from other reviewers, adding visualization experiments of the matching score in the Phi-3 model, experiments using a 2-layer Transformer combined with CoT to achieve multi-step reasoning, and conducted detailed parameter sweeps for the RMBA experiments. We have also rewritten content in the paper that might cause ambiguity, enhancing the rigor of the manuscript. Specific changes can be found in the common response. Your comments, along with those of other reviewers, have greatly helped improve the quality of our paper.
>
> **We look forward to your reply!**

---

> > ### Comment · Reviewer_mhu8 · 2024-11-19
> > **Updates**
> >
> > Thank you for your comments and the changes you made. I have updated my scores according to it.

---

> > > ### Author Response · Authors · 2024-11-19
> > >
> > > We would like to reiterate our sincere gratitude for the recommendations you provided. Moreover, we genuinely appreciate your willingness to raise the score for our work. Should you have any additional questions or comments, please do not hesitate to share them with us.

---

> > > ### Author Response · Authors · 2024-12-02
> > >
> > > Dear reviewer, we would like to once again express our sincere gratitude for your constructive feedback and for recognizing the improvements we made to our manuscript. Over the past two weeks, we have further revised the paper by incorporating both your suggestions and those of the other reviewers. Specifically, we have added **9 pages** of appendices and made detailed updates to the main text. Below is a summary of the key changes we have made:
> > >
> > > > 1. We have added Appendix G to present the detailed inference processes of **Chain-of-Thought (CoT)** reasoning. Additionally, **(new)** we report the accuracy of the model trained with 1-step reasoning data on tasks requiring 1 to 6 steps of reasoning.
> > >
> > > > 2. We have conducted experiments on the **real language model Phi-3**, providing three strong pieces of evidence to support the high likelihood that the buffer mechanism is adopted by large models: (1) the weight alignment phenomenon, (2) **(new)** the reasoning head positions identified using the matching score, which coincide with findings from previous studies, and (3) **(new)** the near orthogonality of the {$W^{vo(l)}$} matrices across layers in Phi-3.
> > >
> > > > 3. We have performed **hyperparameter sweep for the RMBA-related experiments**, conducting a total of 720 experiments to comprehensively validate the role of RMBA in enhancing the reasoning capabilities of models on symbolic tasks.
> > >
> > > > 4. **(new)** We have added Appendix J: Causal Intervention. Through **over 600 causal intervention experiments**, we aim to rule out the possibility that Transformers are using alternative mechanisms when learning from symbolic data.
> > >
> > > > 5. **(new)** We have revised over-claimed sentences in the main text to **enhance the rigor and precision of the paper**.
> > >
> > > We hope that these additional experiments and revisions have further strengthened our paper, and we would greatly appreciate it if you could consider the new results and updates. Your thoughtful feedback has been invaluable to us, and we are confident that these improvements contribute meaningfully to the paper's overall quality. Thank you once again for your time and consideration. We look forward to your continued insights and feedback.

---

> > > > ### Comment · Reviewer_mhu8 · 2024-12-02
> > > >
> > > > I acknowledge the changes in the paper and the additional experiments. I will therefore reconsider my final score accordingly, after a final read of the current version today.

---

> > > > > ### Author Response · Authors · 2024-12-04
> > > > >
> > > > > We extend our sincere gratitude to the reviewers for their support of our work and for taking the additional time to examine the revisions we have made. Your constructive feedback has significantly contributed to enhancing our manuscript. We respect your final evaluation of our article and wish you continued success in your future endeavors.

---

### Official Review · Reviewer_r2jr · 2024-11-01

**Soundness:** 3
**Presentation:** 3
**Contribution:** 3
**Rating:** 8
**Confidence:** 4

**Summary:**

This paper proposes a mechanistic analysis of multi-step reasoning in language models. In particular, the authors design a synthetic task that requires the model to construct an n-step deductive reasoning chain. In this setting, the authors show that a decoder-only single-head transformer implements what they term a “buffer mechanism”: the attention weight matrices W_q and W_k act as information-extractors that retrieve information from a set of buffers implemented by the W_v and W_o matrices. This framework is first formally explained, then empirically verified. The authors then connect this result to the horizontal computation involved in CoT reasoning. Additionally, the paper proposes an approach to modify the attention weight matrices in such a way to create an additional buffer, and presents evidence showing its effectiveness both on the synthetic task designed by the authors and on ProntoQA.

**Strengths:**

- The problem address is Interesting and relevant.
- Solid experimental evidence support the hypothesized mechanism.
- The authors use insights on the model's internal mechanics to propose a method for improving training, which is show to be effective.

**Weaknesses:**

The link to CoT reasoning is somewhat tenuous. Although the authors demonstrate how a 2-layer transformer could theoretically implement a multi-step reasoning algorithm for the synthetic task, this does not imply that a transformer-based language model uses the same approach to perform CoT reasoning. The statement, “in Fig. 6 we illustrate how the Transformer model employs CoT for multi-step reasoning” (lines 341-342), feels overly assertive and lacks supporting empirical evidence.

**Questions:**

Can the authors provide empirical evidence about the hypothesized CoT mechanism?

---

> ### Author Response · Authors · 2024-11-18
>
> Thank you for your constructive comments on our paper. Below are our responses to the issues you raised:
>
> **[R3W1] [Overclaim]** We thank you and the other reviewers for highlighting the overclaim issues. In the revised manuscript, we have addressed these concerns by adding the qualifier *"symbolic multi-step reasoning tasks"* to the relevant statements.
>
> **[R3Q1] [Empirical Evidence]**
>
> 1. In the revised manuscript, we add experiments in the appendix G on how to train a 2-layer model to achieve symbolic multi-step reasoning tasks by performing CoT. In this experiment, we trained a Transformer using only single-step reasoning data. During the testing phase, we allowed the model to use CoT for reasoning by re-inputting the output information from each step back into the model. This enabled us to achieve 2-step and 3-step reasoning. In the appendix, we present the actual information flow diagrams rather than the schematic diagrams shown in the main text. We hope this experiment can supports our viewpoint.
>
> 2. The core idea of this section is that, unlike vertical thinking where the model needs to form multiple buffers, when using CoT, the model repeatedly overwrites the same buffer to achieve multi-step reasoning. Therefore, during the training phase, only a few weight matrices are needed to form the reasoning circuit, which significantly reduces the training difficulty of the model. We have added these points to the main text.
>
> **[Other Revisions]** In addition, we have incorporated suggestions from other reviewers, adding visualizations of the matching scores in the Phi-3 model and conducting a detailed parameter sweep for the RMBA experiments. We have also enriched the explanations of some figures and formulas in the paper, and rewritten content that might cause ambiguity, enhancing the rigor of the manuscript. The specific changes can be found in the common response. Your comments, along with those of other reviewers, have greatly helped improve the quality of our paper.
>
> **We look forward to your reply!**

---

> > ### Comment · Reviewer_r2jr · 2024-11-22
> >
> > Thank you for the response and the additional analysis. I appreciate the additional results about lateral reasoning in Appendix G.
> >
> > > unlike vertical thinking where the model needs to form multiple buffers, when using CoT, the model repeatedly overwrites the same buffer to achieve multi-step reasoning.
> >
> > I see you are mentioning this in Lines 401-402 too. Can you clarify what evidence are you using to support this claim? My understanding is that we can conclude this because a model trained to perform on 13-step is able to correctly predict the next tokens after the 13th token (Figure 18). I think this should be made clear in the main text.
> >
> > Also, can the results illustrated in Figure 18 be quantified? E.g., with what accuracy does the model predict the tokens after the 13? Are the attention scores represented by the thickness of the lines average over multiple examples or are they for a single example?
> >
> > Minor point: The tile of Appendix G reads "Detials" instead of "Details"

---

> > > ### Author Response · Authors · 2024-11-23
> > >
> > > We appreciate the reviewer's thorough examination of the numerous changes we have made and for providing further suggestions. Below are our detailed responses to each of your recommendations; the corresponding revisions are highlighted in blue in the new manuscript.
> > >
> > > **[Evidence to support claim]** The inner mechanism of CoT is actually identical to the mechanism of the first reasoning step in the vertical thinking strategy. In the new manuscript, we have added Figure of the CoT matching matrix in Appendix G to make our conclusions more convincing. In the main text, we have incorporated the content you mentioned about length generalization. Specifically, we have added the following(bold part):
> > >
> > > >*We trained a 2-layer Transformer with the **13-length** single-step reasoning data. During the testing phase, we fed the model's output back into the model. Through this CoT process, the model can perform 2-step, 3-step, or even higher-step reasoning, **and it can also generalize to sentence lengths beyond the 13th position.***
> > >
> > > **[Quantify Figure 18]** This is an excellent suggestion. In Appendix G of the new manuscript, we have **added a figure (Figure 20) to quantify the accuracy of the network trained with single-step reasoning data when using CoT to perform multi-step reasoning.** The new results show that when using CoT, the model achieves an accuracy of 84.1% on 3-step reasoning tasks. Even for the most complex 6-step reasoning, it achieves an accuracy of 57.6%.
> > >
> > > **[Attention scores in Figure 18]** The thickness of attention lines shown in Figure is for a single example.
> > >
> > > **Undoubtedly, your revision suggestions during this period have greatly helped us enrich the content of the CoT section. We look forward to your reply again!**

---

> > > ### Author Response · Authors · 2024-11-27
> > >
> > > Dear reviewer, we would like to thank you again for your valuable feedback during the review process. Today is the final deadline for submitting the revised manuscript, and we are concerned that you may not have noticed the changes we made in response to your previous comments.
> > > Over the past period, we have engaged in discussions with other reviewers, and based on your and their suggestions, we have made further revisions and expansions to the manuscript. Specifically, we have added **9 pages** of appendix and made detailed updates to the main text. Here is a brief summary of the key changes we made:
> > >
> > > > 1.We have added Appendix G to present the detailed inference processes of Chain-of-Thought (CoT) reasoning. Additionally, **we report the accuracy of the model trained with 1-step reasoning data on tasks requiring 1 to 6 steps of reasoning.**
> > >
> > > > 2.We have conducted experiments on the **real language model Phi-3**, providing three strong pieces of evidence to support the high likelihood that the buffer mechanism is adopted by large models: (1) the weight alignment phenomenon, (2) the reasoning head positions identified using the matching score, which coincide with findings from previous studies, and (3) the near orthogonality of the {$W^{vo(l)}$} matrices across layers in Phi-3.
> > >
> > > > 3.We have performed **hyperparameter sweep for the RMBA-related experiments**, conducting a total of 720 experiments to comprehensively validate the role of RMBA in enhancing the reasoning capabilities of models on symbolic tasks.
> > >
> > > > 4.**(new)** We have added Appendix J: Causal Intervention. Through **over 600 causal intervention experiments**, we aim to rule out the possibility that Transformers are using alternative mechanisms when learning from symbolic data.
> > >
> > > > 5.**(new)** We have revised over-claimed sentences in the main text to **enhance the rigor and precision of the paper**.
> > >
> > > In addition, the other three reviewers have given positive feedback on the improvements we made and have accordingly increased their ratings. **We sincerely hope that you can further support our work and would appreciate it if you could let us know whether you are satisfied with our responses to your previous questions.** If you have any additional concerns or suggestions, we would be more than happy to discuss and make any further adjustments. Thank you once again for your time and thoughtful feedback.

---

> > > > ### Comment · Reviewer_r2jr · 2024-11-29
> > > >
> > > > I think the additional experiments and analyses carried out by the authors made the paper a stronger submission. I have raised my overall rating to reflect these improvements.

---

> > > > > ### Author Response · Authors · 2024-11-30
> > > > >
> > > > > We would like to express our sincere gratitude for your support of our work. We also appreciate your constructive suggestions for improving our manuscript. We are pleased to hear that the additional experiments and analyses we recently conducted have enhanced the overall quality of the paper. We are deeply thankful for your willingness to consider increasing the score for our work.

---

### Official Review · Reviewer_JU6V · 2024-11-02

**Soundness:** 3
**Presentation:** 4
**Contribution:** 3
**Rating:** 6
**Confidence:** 3

**Summary:**

The paper proposes a "Buffer Mechanism" that could be used by Transformers to perform reasoning tasks.  For a specific synthetic reasoning task, the authors give a concrete mathematical description of how the proposed Buffer Mechanisms might implement vertical-style reasoning (through the layers of a Transformer), and horizontal-style reasoning (like Chain-of-Thought).  The authors also suggest a tweak (arising from their Buffer Mechanism formulation) by which the performance of Transformers might be enhanced.

**Strengths:**

The proposed Buffer Mechanism makes sense, though its description seems a little obfuscated.  The paper sets an ambitious and laudable goal : Understanding of how Transformers can implement the learning of reasoning.  The experiments included demonstrate that, when measured using benchmarks chosen by the authors, the method suggested is effective.  The performance-enhancing tweak suggested makes sense in the context of making reasoning mechanisms more learnable overall - certainly an interesting direction.

**Weaknesses:**

The paper make the bold assumption that the "Buffer Mechanism" that is described is the specific mechanism that Transformers are using to perform reasoning.  Before being proven, this seems like it should be treated more like a hypothesis.  And the experiments given do not prove anything in general, rather point towards this mechanism being plausible for their particular set-up.  For example:
* L079 : "Specifically, we found that Transformers utilize a Buffer Mechanism when engaging in complex reasoning".  As far as this reviewer can ascertain, this was not proven, even experimentally.
* L091 : "We discover the buffer mechanism employed by language models" - again this is a very broad claim, given the evidence shown.
* etc...

So perhaps, it would be better to restate quite a few of the claims made...
* L501 : "we investigated the buffer mechanism employed by Transformer models" -> "we investigated how Transformers might implement a buffer mechanism"

In the section around L248 : "This indicates that the model has learned the underlying patterns in the data."  This seems like a statement about a property of model training, whereas the result indicated by Eqn 11 depends on L238 ("When same-token matching happens") - which appears to be circular logic.  (i.e. given that it has learned X, then it has learned X')

Minor things:
* Figure 1 : Testing each LLM on the same question 9 times, and listing out the results (rather than, for instance, showing a histogram) really emphasised how little the performance of these LLMs was investigated
* Figure 2 : "Serialized Reprentation" -> "Serialized Representation"

**Questions:**

Would it be reasonable to have a mental model of the Buffer Mechanism as being the result of there being a basis (~L209) in which the layer-wise transformations/matching for the Transformer internal representations are approximately block-diagonal (these sub-representations being the buffers)?

Figure 5a : Why does the test accuracy go down before epoch 50?  And then leap up?  What happens at epoch 50?

How are the embeddings of the tokens learned (in particular the OOD tokens)?

Doesn't Appendix D indicate that "The Buffer Mechanism" presented in the body of the paper is likely only a partial explanation of what is actually being performed by Transformers?

---

> ### Author Response · Authors · 2024-11-18
>
> Thank you for the constructive feedback on our manuscript. Below are our responses to each of your comments:
>
> **[R2W1] [Overclaim]** We appreciate the reviewer pointing out the overclaim issues in our writing. We have revised all instances of overclaim identified in our manuscript. Specifically, we have added the qualifier *"symbolic multi-step reasoning tasks"* to each relevant claim.
>
> **[R2W2] [Circular Logic]**
> The ambiguity in this section was due to our inadequate phrasing; in fact, there is no circular reasoning involved. We have revised the relevant section in the new manuscript as follows:
> *To achieve the same-token matching, it is sufficient for $\text{Ker}^{(l)} \approx I$, in which case*
> $$h^{(l)}(X_{tgt}) \approx X_{tgt}X_{tgt}^T = I + O(1/\sqrt{d_m}).$$
> *This equation indicates that the attention of all tokens is focused on themselves. Furthermore, we observe that same-token matching is independent of the specific value of $X_{{tgt}}$. For example, for $X_{{tgt, OOD}}$ sampled from untrained random vectors $token_{OOD}$, $h^{(l)}(X_{{tgt, OOD}}) \approx I$ still holds. Therefore, when the model weights satisfy $Ker^{(l)} \approx I$, the model demonstrates out-of-distribution generalization capability.*
>
> **[R2W3] [Minor Things]**
> We conducted more detailed interactions with various large models. Specifically, we used four GPT and Claude models to perform 2-, 3-, and 4-step reasoning tasks, examining their performance with and without Chain-of-Thought (CoT) prompting. Each setup was repeated 25 times, **totaling 600 interactions**, to ensure statistically robust results. The findings show that current large models struggle to perform 2(or higher)-step reasoning tasks directly. However, with CoT prompting, all these models excel at multi-step reasoning. Based on these results, we have revised Figure 1 and included detailed interaction results in the Appendix. Additionally, we have corrected the typographical error in Figure 2.

---

> ### Author Response · Authors · 2024-11-18
>
> **[R2Q1] [Reasonable]**
> There is an intuitive explanation for the diagonal structure observed between the $W^{vo}$ matrix of the previous layer and the $W^{qk}$ matrix of the next layer: information derived in one layer is directly utilized by the next layer, aligning with logical reasoning.
>
> 1. **Diagonal Structure in Real Models**:
>    In the revised manuscript, we have supplemented experiments using the Phi-3 model to compute matching scores and kernel scores, and defined methods for calculating these metrics in multi-head models. Key observations include:
>    - **(a)** When $l_1 \leq l_2$, $W^{qk(l_1)} W^{vo(l_2),T}$ exhibits almost no significant features (kernel score ≈ 0).
>    - **(b)** When $l_1 > l_2$, the kernel score of $W^{qk(l_1)} W^{vo(l_2),T}$ decreases as $l_1 - l_2$ increases. These observations indicate that information stored in earlier layers (by buffer $W^{vo(l_2)}$) is extracted by later layers (via $ W^{qk(l_1)}$), and the model tends to extract the most recently processed information.
>    - **(c)** Heads with high matching scores are mostly concentrated in the middle layers, which aligns with previous findings that reasoning in large models predominantly occurs in the middle layers.
>
> 2. **Improving Reasoning**:
>    Both the RMBA algorithm proposed in our paper and the IMBA algorithm by Boix-Adsera E, et al.[1] improve the model's reasoning capability. This supports the conclusion that the diagonal structure of $W^{qk(l_1)}W^{vo(l_2),T}$ facilitates information transfer.
>    *Reference*: [1] Boix-Adsera E, et al., *When can transformers reason with abstract symbols*
>
> 3. **Future Directions**:
>    The real-world data have different characteristics (e.g., memory-centric vs. reasoning-centric). In the future, the MOE (Mixture of Experts) architectures could potentially be designed to incorporate diagonal structures in certain layers and heads, improving both reasoning and memory capabilities.
>
> **[R2Q2] [Accuracy Suddenly Improved]**
> This is a good question. Understanding the dynamics of how models learn multi-step reasoning is a key direction of our future work. Currently, we lack a rigorous explanation but have observed that this phenomenon is consistent, regardless of changes to learning rate, weight decay, $d_m$, and $d_k$. When the number of epochs is fewer than 50, the model has yet to establish a stable information flow (even on the training data) and appears to rely more on memorization. The cause of this phenomenon remains unclear.
>
> **[R2Q3] [OOD Embedding]**
> In fact, Transformers do not need to learn embeddings for OOD tokens. Based on our mechanism analysis and random matrix theory, as long as $Ker = I$, it follows that $h(X_{{OOD}}) \approx X_{{OOD}}X_{{OOD}}^T \approx I$, ensuring same-token matching. This is one of the most intriguing aspects of the paper, as the OOD generalization capability effectively characterizes whether the model has truly “understood” the rules.
>
> **[R2Q4] [Parallel Reasoning]**
> The parallel reasoning described in Appendix also utilizes the buffer mechanism for information transfer. Here, the buffer mechanism is represented as {$\{\prod_{l \in J} \mathbf{W}^{vo(l)} | J \subset \{0, 1, \ldots, L-1\}\}$}. Figure 12(c) shows that the model still stores different information in different buffers. This section complements the vertical reasoning section and applies to cases where the reasoning steps $n \geq L$ (model depth). It does not conflict with the main text, which focuses on $n < L$. Generally, the depth of a large language model greatly exceeds the reasoning steps required for single predictions, especially with the prevalent use of CoT prompting in modern large models.
>
> **[Additional Revisions]**
> In addition, we have incorporated feedback from other reviewers by adding experimental studies on CoT and conducting a thorough parameter sweep for the RMBA experiments. We also enriched the explanations of some figures and formulas and clarified ambiguous content to improve the rigor of the paper. Detailed revisions can be found in the common response. Your comments, along with those of other reviewers, have greatly helped improve the quality of our paper.
>
> **We look forward to your reply!**

---

> > ### Comment · Reviewer_JU6V · 2024-11-24
> >
> > [Re: R2W1] [Overclaim]
> >
> > Perhaps it wasn't clear (and another reviewer picked up on the same thing) : The paper is proposing *A* Buffer Mechanism, and appears to find that such a mechanism can be detected when it is looked for.  But the paper doesn't show that transformers are solely using *THE* Buffer Mechanism that you are proposing. There could be many mechanisms, but they were not looked for here.  Certainly, if you looked for a Buffer Mechanism and didn't find any evidence, then that proposal wouldn't even be in a paper.  Here, you did find what you were looking for, but that is not proof that you have discovered THE Buffer Mechanism used by Transformers, nor that other mechanisms aren't being used as well.
> >
> > Your update (restricting the claim to "symbolic multi-step reasoning task") doesn't tackle the heart of the issue (which is present in the title of the paper, so it may be be difficult to change course at this point).
> >
> >
> > [Re: R2W3] [Minor Things]
> >
> > Good update, which supports the idea that large-models require CoT much more clearly.
> >
> >
> > [Re: Diagonal Structure in Real Models]
> >
> > Good to see that the same Buffer Mechanism can be seen in a real model (Phi-3).
> >
> > But again, finding something that you were looking for is only evidence that your claims might be true (your claim in the Introduction is "We discover the buffer mechanism employed by language models during the reasoning process in symbolic multi-step reasoning tasks").  It seems like "We found evidence that supports a buffer mechanism (which we describe herein) being employed by language models during the reasoning process in symbolic multi-step reasoning tasks" would be better supported.
> >
> >
> > [Re: R2Q2] [Accuracy Suddenly Improved]
> >
> > This reviewer finds it odd that epoch=50 is such a turning point.  In (for instance) grokking behaviours, the turning points are spread over an order of magnitude (or more).  So the sudden take-off is remarkable.
> >
> >
> > [Re: R2Q3] [OOD Embedding]
> >
> > Now this part seems like a much stronger argument in favour of a buffer mechanism being a prime-candidate as an explanation (because there are dramatically fewer mechanisms that could make use of untrained embeddings, making your Buffer Mechanism a stronger candidate).  It should definitely be made clearer that the OOD embeddings are not trained (i.e. left at random initialisation).
> >
> >
> > [Re: R2Q4] [Parallel Reasoning]
> >
> > Again, not being in conflict with the main text is not the same as being strong evidence for the theory.
> >
> >
> > [Overall:]
> >
> > My ratings are unchanged : I'm relucant to increase (despite the obvious hard work the authors have performed), since the main claim is a short distance from a presentation that starts "We discovered how transformers do reasoning!"

---

> > > ### Author Response · Authors · 2024-11-24
> > >
> > > We appreciate the reviewers for their thorough review of all the comments and suggestions, as well as the detailed feedback on our previous revisions. Below, we address each of your comments individually. Corresponding second-round modifications are highlighted in blue in the revised manuscript.
> > >
> > > **[Overclaim]**
> > >
> > > We agree that our earlier revisions did not fully address the issue of overclaiming. Considering the complexity of language tasks and language models, defining any mechanism clearly within real-world large language models is nearly impossible. Our initial intent was:
> > >
> > > (1)To identify or propose mechanisms that real language models might employ.
> > >
> > > (2)To validate that this mechanism is indeed employed by the model in certain simple tasks.
> > >
> > > To address this more rigorously, we made the following additional revisions:
> > >
> > > (a)The title of the paper has been updated to: **A Buffer Mechanism Employed in Language Models to Implement Symbolic Multi-Step Reasoning Tasks** (It’s allowed by policy)
> > >
> > > (b)We revised the statement:
> > >
> > > > We discover the buffer mechanism employed by language models during the reasoning process in symbolic multi-step reasoning tasks
> > >
> > > to:
> > >
> > > > We found evidence that supports a buffer mechanism being employed by language models during the reasoning process in symbolic multi-step reasoning tasks.
> > >
> > > (c)As described in our second-round response to Reviewer [fnjG], we conducted a series of causal intervention experiments to show that, in a well-trained 3-layer model, altering the information stored in the final token's buffers results in reasoning failure. We believe this task demonstrates that the model employs a buffer mechanism for reasoning in symbolic tasks. If other compensatory mechanisms exist, the model should, at least once, be able to produce correct outputs after critical information for the buffer mechanism has been disrupted. However, this was not observed in our results. The experimental details and methodology can be found in Appendix J of the revised manuscript.
> > >
> > > **[Accuracy Suddenly Improved]**
> > >
> > > This phenomenon may differ from the previously studied grokking phenomenon. It could represent another significant behavior, and we plan to focus on the model's dynamics in subsequent research to investigate this further.
> > >
> > > **[OOD Embedding]**
> > >
> > > We have added the following explanation to the main text to highlight the principle behind OOD generalization:
> > >
> > > > Furthermore, a much more remarkable observation is that the same-token matching is independent of the specific value of $X_{tgt}$. For example, for $X_{tgt,OOD}$ sampled from the **untrained random vectors** $token_{OOD}$, $h^{(l)}(X_{tgt,OOD}) \approx I$ still holds. Therefore, when the model's weights satisfy $Ker^{(l)} \approx I$, the model exhibits out-of-distribution generalization capability.
> > >
> > > **[Parallel Reasoning]**
> > >
> > > We agree with the reviewer’s perspective. This section is just a supplement to the vertical reasoning discussion in the main text. In the revised manuscript, we have rephrased the buffer mechanism in the Parallel Reasoning section as "a possible explanation." We hope this section serves as an open-ended exploration that enriches the article's content and inspires future work, while avoiding overclaiming.

---

> > > > ### Author Response · Authors · 2024-11-24
> > > >
> > > > **[Contribution]**
> > > >
> > > > In our initial draft, we unintentionally made several overclaims. With the assistance of the reviewers, we believe the rigor of our manuscript has significantly improved, and we are committed to addressing any further issues of rigor that may remain.
> > > >
> > > > Our work is not centered on the claim: “We discovered how transformers do reasoning!” Rather, our goal is to present evidence of a mechanism that Transformers might use for reasoning. (Interestingly, our original title was “Toward Understanding How Transformers Deal with Multi-Step Reasoning”. While we changed the title to avoid overclaiming, we still overlooked the distinction between “The” and “A” in English.)
> > > >
> > > > Beyond the issue of overclaiming, we would like to restate the key contributions of our work:
> > > >
> > > > (1)We propose a potential buffer mechanism and provide theoretical characterization and experimental analysis of this mechanism on a symbolic dataset.
> > > >
> > > > (2)Using the buffer mechanism and symbolic dataset, we differentiate between vertical and horizontal reasoning strategies. From the perspective of buffers, we offer a plausible explanation for why large models with CoT strategies often excel in reasoning tasks.
> > > >
> > > > (3)We utilize the buffer mechanism to explain why Transformers exhibit OOD token generalization capabilities and why IMBA fails on multi-step reasoning datasets.
> > > >
> > > > (4)With the understanding of buffer mechanism, we propose the RMBA, which significantly improves learning efficiency on the PrOntoQA task.
> > > >
> > > > (5)In the Phi-3 model, we observe weight alignment phenomena that seem to support the existence of the buffer mechanism.
> > > >
> > > > We are sincerely grateful for the reviewer’s suggestions regarding our overclaim issues. We firmly believe that through our collaborative efforts, this article has been greatly enhanced in value. **Given that we have carefully revised all known overclaim issues, we would deeply appreciate it if the reviewer could reassess the substantive contributions of our work. Your evaluation is truly important to us!**

---

> > > > > ### Comment · Reviewer_JU6V · 2024-11-26
> > > > >
> > > > > I would have taken a different approach with "Toward(s) Understanding How Transformers Deal with Multi-Step Reasoning"...
> > > > >
> > > > > I've increased my score by a notch.

---

> > > > > > ### Author Response · Authors · 2024-11-27
> > > > > >
> > > > > > We greatly appreciate the reviewers' recognition of our work and the recent revisions made! Your numerous suggestions regarding rigor have significantly enhanced the quality of this article. Moving forward, we will continue to prioritize the rigorous articulation of our arguments. Should you have any further questions, please feel free to contact us.

---

### Official Review · Reviewer_fnjG · 2024-11-03

**Soundness:** 3
**Presentation:** 3
**Contribution:** 2
**Rating:** 5
**Confidence:** 4

**Summary:**

The paper studies the 'buffer mechanism' in transformers for solving a in-context multi-hop reasoning task. The task requires chaining associations present in the context (e.g. "a->b, b->c") to perform multi-hop reasoning (e.g. "a->c"). Inspired by the copying + induction head mechanisms found in prior work, the authors hypothesize that transformers trained to perform this task sequentially look up the necessary reasoning steps and form intermediate answers at the last token position, so that the second layer looks up the first hop, the third layer looks up the second hop, etc. By default the intermediate representations persist in the residual stream at the last token; to prevent the intermediate representations for one hop from interfering with those for a later hop, the authors hypothesize that the intermediate representations are kept in separate 'buffers', i.e. linearly isomorphic subspaces that are disjoint from each other.

The authors empirically test the hypothesis by measuring the alignment of certain weight matrices in a 3 layer transformer trained to do 2-hop reasoning. Further, the authors proposed a parametrization of the attention weight matrices that biases the network towards forming these disjoint subspaces that reduce interference. They find that this parametrization reduces the number of training steps needed for a 12-layer transformer to learn the ProntoQA task.

**Strengths:**

- Multi-hop reasoning in language models is an important problem to study
- The figures are beautifully made
- The observation that multihop reasoning can generalize to unseen tokens because the embeddings at initialization already have the required properties is neat

**Weaknesses:**

- The paper centers its analyses on small transformers trained on synthetic tasks. It is not clear if any finding can be transferred to practical language models trained on text. Prior work has already studied two-hop reasoning in practical language models in more realistic tasks --- a more convincing demonstration of the buffer mechanism is to identify it in these settings.

Overall, the empirical analyses are weak and can benefit from more careful experimental techniques. In particular:
- In sec 3.1, the authors argue that a particular transformer that they trained exhibit the buffer mechanism by comparing metrics about how aligned certain weight matrices are. This is much weaker and more circumstantial than directly performing causal interventions on the activations to verify their claims.
- Further, the authors did not empirically compare the buffer mechanism with any competing mechanisms. Off the top of my head, I imagine one competing explanation would be that instead of storing intermediate representations in distinct buffers, maybe the transformer instead learns to overwrite stale representations with new representations. Such a mechanism could conceivably still possess similar weight alignment patterns as the buffer mechanism.
- Exacerbating this, the authors only performed this analysis on a 3-layer transformer, whereas their diagram (Fig. 3) and their theoretical construction would work for arbitrarily deep models. It is conceivable that interesting divergences between the buffer mechanism and the overwriting mechanism may only appear at deep enough models.

**Questions:**

- I can understand the idea behind matching score, but can you provide motivation for the kernel score? Why is there a factor of $n$ in the numerator? My understanding is that Ker should be a $d_m \times d_m$ matrix, so it's mystifying to me why $n$ is there. It's also really hard to interpret this quantity. In comparison, I think your matching score is makes a lot more sense, because the softmax constrains it to [0,1], and so I know how to interpret values. My personal guess is that operator norm of the difference between the kernel and identity would be a good metric, but I'd be curious to hear the reasoning behind your choice.
- Do you perform hyperparameter sweeps for the results in sec. 5? If not, the finding that RMBA speeds up training could simply be because the parametrization some how matches the default hyperparameters you use, and not because of any real gains. I believe at the very least you should sweep learning rate for each of the 3 methods and record the best result.

---

> ### Author Response · Authors · 2024-11-18
>
> Thank you for your constructive comments on our paper. Below are our detailed responses to each of your points:
>
> **[R1W1.1] [Synthetic Tasks]**
>
> 1. **[Why start from synthetic tasks?]** Due to the complexity of linguistic data and large language models, it is challenging to identify key mechanisms through direct research on real language data, limiting us to relatively macro-level statistical analyses. We chose to conduct research on simplified symbolic datasets, which allows us to perform sufficient numerical experiments with minimal computational resources and to closely observe the internal workings of the Transformer model. The buffer mechanism we proposed aids in understanding the expressive capabilities of Transformer models.
>
> 2. **Applicability to real large language models.** In the revised manuscript, we have added experiments calculating the matching score and kernel score in the Phi-3 model. We provided methods for computing the matching score and kernel score in multi-head models. By calculating the kernel score, we assessed the strength of inter-layer information interactions in Phi-3 and computed the matching score for each head in each layer. We found:
>
>    - **(a)** When $l_1 \leq l_2$, $W^{qk(l_1)} W^{vo(l_2),T}$ exhibits almost no significant features (kernel score ≈ 0).
>    - **(b)** When $l_1 > l_2$, the kernel score of $W^{qk(l_1)} W^{vo(l_2),T}$ decreases as $l_1 - l_2$ increases. These observations indicate that information stored in earlier layers (by buffer $W^{vo(l_2)}$) is extracted by later layers (via $W^{qk(l_1)}$), and the model tends to extract the most recently processed information.
>    - **(c)** Heads with high matching scores are mostly concentrated in the middle layers, which aligns with previous findings that reasoning in large models predominantly occurs in the middle layers.
>
> 3. **Algorithmic insights from synthetic tasks.** The buffer mechanism we discovered based on synthetic tasks can inspire algorithm design. For example, as shown in Figure 8, the RMBA algorithm proposed—motivated by the buffer mechanism—can also aid in training real large language models.
>
> **[R1W1.2] [Prior Work]**
>
> According to our literature review, most of the current work in the field on multi-hop reasoning are evaluations of existing large models or use causal intervention starting from results to identify key pathways. In the *Related Work* section of the revised manuscript, we have added references to several recent articles related to multi-hop reasoning and causal intervention and discussed the novelty of our work. If the reviewer could provide more relevant references, we can more specifically discuss the connections and differences between our work and these studies.
>
> **[R1W2] [Causal Intervention]**
>
> 1. **Complementarity with our approach.** First, there is no contradiction between these two studies, as causal intervention is one method to find key information transmission paths, while the buffer mechanism attempts to explain how this information transmission occurs. Moreover, causal intervention can only identify key paths but cannot fundamentally understand how these paths are generated, offering limited guidance for algorithm design. Our paper aims to study what kinds of model weights cause these paths to emerge, providing a more detailed modeling of the implementation mechanisms of causal paths. Accordingly, the proposed RMBA can also promote the generation of reasoning logic circuits.
>
> 2. **Progress through synthetic tasks and aligned weight matrices.** Some existing works have made significant progress via "synthetic tasks" and "weight alignment." For example, Boix-Adsera E, et al. [1] proposed the IMBA by studying single-step reasoning, which has been proven to aid in learning simple reasoning data. However, from the buffer perspective, we found that this algorithm causes the buffer to degenerate, making it difficult to learn multi-step reasoning. Therefore, we propose the RMBA to further improve the model's accuracy in multi-step reasoning tasks.
>
>    [1] Boix-Adsera E, et al. *When can transformers reason with abstract symbols*
>
> 3. In the field of biological neural networks, performing causal intervention is common because the methods for recognizing and processing biological signals are limited. However, in artificial neural networks, we have richer means to recognize and process signals, allowing us to obtain deeper insights than those in biological neural network research.

---

> > ### Author Response · Authors · 2024-11-18
> >
> > **[R1W3] [Competing Mechanisms]**
> >
> > 1. **Existence of the overwrite buffer mechanism.** The overwrite buffer mechanism you mentioned does exist. In Section 4 of our paper, the Chain-of-Thought (CoT) method conduct multi-step reasoning by overwriting the same buffer, which explains why CoT can enhance model reasoning (because it only requires aligning 2 pair of weight matrices).
> >
> > 2. **Non-existence in vertical thinking frameworks.** However, for the vertical thinking framework, the overwrite buffer mechanism does not exist because the weight matrices (buffer) $W^{vo(l)}$ of each layer are different.
> >
> > **[R1W4] [3-Layer vs. Arbitrarily Deep Models]**
> >
> > The buffer mechanism was inspired by our observations in 3-layer models. In fact, we have also conducted numerical experiments on deeper models (e.g., 4-layer models performing 3-step reasoning), and the conclusions are consistent. The theoretical model we constructed (Eq. 14 & 15) is applicable to models with an arbitrary number of layers. Figure 3 is an embellished schematic diagram; The actual information flow diagrams derived from the theoretical model can be found in Appendix D3.
> >
> > Additionally, in the appendix, we have added experiments training a 2-layer Transformer using 1-step reasoning data, where the information flow is the same as what we presented in the main text. We also demonstrated that this network, combined with the CoT strategy, can perform higher-step reasoning, such as 3-step reasoning. These experiments further illustrate the generality of the buffer mechanism.
> >
> > **[R1Q1] [Kernel Score]**
> >
> > There was a typo in the original manuscript regarding the definition of the kernel score. Our original intention was to define Kernel Score (Ker) = mean(|$Ker_{ii}$|) / mean(|$Ker_{ij}$|). Thank you for pointing this out. In the revised manuscript, we have unified the definitions of the kernel score and matching score. We define the kernel score as Trace(softmax(Ker)) / $d_m$. Kernel = I provides the intrinsic driving force for the matching matrix $h(X)=I$ in out-of-distribution (OOD) tokens. If we do not consider nonlinear effects such as LayerNorm and feed-forward networks (FNN), the kernel matrix simplifies the study. However, since the matching score is directly related to which tokens the model focuses on and can consider nonlinear effects, it is indeed a more meaningful metric in practical applications.
> >
> > **[R1Q2] [Hyperparameter Sweep for RMBA]**
> >
> > To make the results more convincing, we have added in the appendix of the revised manuscript the impact of different hyperparameters on the results of these three methods. We selected 4 weight decay values, 5 learning rates, and 4 configurations of hidden space dimensions. Under each setting, we conducted 3 experiments for each of the 3 algorithms (baseline, IMBA, RMBA), **totaling 720 experiments**. In most cases, RMBA achieved better results than the other algorithms.
> >
> > **[Other Revisions]**
> >
> > In addition, we have incorporated the suggestions of other reviewers to enrich the explanations of some figures and formulas in the paper. We have rewritten content that might cause ambiguity, enhancing the rigor of the manuscript. The specific changes can be found in the common response. Your comments, along with those of other reviewers, have greatly helped improve the quality of our paper.
> >
> > **We look forward to your reply!**

---

> > > ### Comment · Reviewer_fnjG · 2024-11-23
> > >
> > > __Hyperparameter sweep__
> > >
> > > Thanks for doing the experiments. It looks like for the optimal setting (weight decay=0.1, learning rate = 0.0001), all three parameterizations can sometimes achieve the same accuracy at the same time. Is the interpretation then that the RMBA parameterization can more robustly reach a certain accuracy by a certain number of training steps across a wider range of hyperparameters?
> > >
> > > __Competing mechanisms__
> > >
> > > I'm not convinced that your experiments show that the overwrite mechanism doesn't exist in the vertical thinking framework. Can you specify how you think the kernel or matching scores will be different for the overwrite vs buffer mechanism, and show evidence for one mechanism over the other?
> > >
> > > __Application to Phi-3 model__
> > >
> > > I understand that you have plotted the kernel scores for Phi-3, which essentially measures some form of alignment between the weight matrices from attention heads in one layer to another. I understand that you found that this alignment for $W^{QK}W^{VO\top}$ only exists if $W^{QK}$ comes from a later layer than $W^{VO}$. Could you explain how this is evidence for the buffer mechanism per se, and not an example of the many weight alignment phenomena found? For example, the original induction head paper also found alignment between copying and induction heads (Elhage et al, 2021). I'm guessing one part of the evidence is to show that $W^{VO}$ and $W^{QK}$ are individually not diagonal, but their composition is.
> > >
> > > __General comment about experimental rigor__
> > >
> > > Overall, I'm not convinced that enough rigor has gone into experiment design. I'm happy to see that the authors take steps to improve the paper in this regard with the hyperparameter sweep, but I believe more should be done. Specifically, the main evidence presented (kernel score and matching score) are implications of the buffer mechanisms, but are insufficient for showing their existence. If the authors can provide coherent and definitive experiments that rule out plausible mechanisms (overwrite, ordinary induction heads), I will raise my score accordingly.
> > >
> > >
> > > Aubry, Murdock, et al. "Transformer Block Coupling and its Correlation with Generalization in LLMs." arXiv preprint arXiv:2407.07810 (2024).
> > >
> > > Elhage, Nelson, et al. "A mathematical framework for transformer circuits." Transformer Circuits Thread 1.1 (2021): 12.

---

> > > > ### Author Response · Authors · 2024-11-24
> > > >
> > > > We appreciate the reviewers for carefully reviewing our numerous revisions and providing further valuable suggestions. Below, we address your comments one by one. Corresponding second-round modifications are highlighted in the revised manuscript using blue text.
> > > >
> > > > [Hyperparameter Sweep]
> > > > We assume the reviewer is referring to experiments conducted under the configuration of $d_m=256, d_k=128$. Indeed, as shown in the figure, both the baseline and the IMBA method are highly sensitive to random seeds (primarily affecting the initialization of weights). Among three trials with different random seeds, only one accuracy curve aligns with the accuracy curve corresponding to RMBA.
> > > >
> > > > From a rigor perspective, we agree that the statement in the manuscript,
> > > > >"The findings indicate that under these configurations, the RMBA consistently improves the accuracy on this reasoning task,"
> > > >
> > > > should be revised to:
> > > > >"The findings indicate that RMBA parameterization can more robustly reach a high accuracy by a certain number of training steps across a wider range of hyperparameters."
> > > >
> > > > [Competing Mechanisms]
> > > >
> > > > 1. Overwrite Mechanism
> > > >
> > > > To ensure clarity and avoid ambiguity, we enumerate two overwrite mechanisms we can conceive of:
> > > >
> > > > (1) For the same token, the information [a] stored in the buffer $W^{vo(l_0)}$ is replaced by new information [b] generated in layer $l_0+1$ (or subsequent layers).
> > > >
> > > > (2) For two different tokens (denoted as token 1 and token 2), the buffer $W^{vo(l)}$ stores information [a] and [b] respectively. Similar to CoT strategies, the model uses information [a] from token 1 during the first reasoning step and information [b] from token 2 during the second reasoning step. Both reasoning steps utilize information from the same buffer $W^{vo(l)}$.
> > > >
> > > > For overwrite mechanism (1), theoretically, this situation is impossible. Each layer has a new $W^{vo(l)}$, and due to the inherent properties of the Transformer architecture, new information [b] from layer $l_0+1$ will be passed in the form of [b]$W^{vo(l_0+1)}$, not [b]$W^{vo(l_0)}$. However, in practice, special cases may arise where $W^{vo(l)}=W^{vo(l_0+1)}$. A simple validation method is to compute the cosine similarity of $W^{vo(l)}$. In the revised manuscript, we include heatmaps of cosine similarities between $W^{vo(0)}, W^{vo(1)}, W^{vo(2)}, I$ in Appendix D.3 to confirm that they represent four distinct buffers rather than a single one.
> > > >
> > > > For overwrite mechanism (2), we have added Appendix J to present the causal intervention experiments. In contrast to previous causal intervention methods that replace all information within a token, we narrowed the scope to focus on a specific buffer within the token. First, we identified critical tokens and logical circuits through masking the information transfer paths. Then, we selectively replaced information in each buffer of the critical token to observe the model’s output. Results show that in layer $l$, information in the buffer $W^{vo(l-1)}$ of the final token is crucial, whereas altering information in (final token’s) other buffers does not affect the model's output. Coupled with the near-zero cosine similarity between $W^{vo(0)}, W^{vo(1)}, W^{vo(2)}, I$ mentioned above, we can assert that the model utilizes distinct buffers rather than overwriting the same buffer.
> > > >
> > > > 2. Induction Heads
> > > >
> > > > The buffer mechanism proposed in this work can be regarded as an extension and supplement to the induction head mechanism. From a functional perspective, "same token matching" mentioned in this work is similar to induction heads. However, a key objective of this work is to demonstrate that **multi-step reasoning tasks are not equivalent to performing independent single-step reasoning multiple times.** This distinction highlights the research value of studying multi-step reasoning. To this end, we provide the following evidence (focusing on symbolic multi-step reasoning tasks):
> > > >
> > > > (1)From the perspective of mechanism analysis, based on Figures 3 and 4 and the new added causal intervention experiments, the model always stores new reasoning results in a new buffer and then performs same-token matching. The multi-step reasoning mechanism can be seen as a generalization of induction heads. Unlike "independent induction head usage," we require buffer differentiation across layers, which has not been discussed in prior work.

---

> > > > > ### Author Response · Authors · 2024-11-24
> > > > >
> > > > > (2)From a numerical simulation perspective, although there is theoretical and numerical  evidence that IMBA promotes the formation of induction heads, applying IMBA to 2-step reasoning tasks shows that it fails to improve model accuracy effectively in most settings. This aligns with our buffer theory, as IMBA sacrifices buffer differentiation. This direct evidence is summarized in Section 5 of the main text:
> > > > >
> > > > > > These results also demonstrate that multi-step reasoning is not achieved by simply ``stacking" multiple single-step reasonings. An algorithm that enhances single-step reasoning may not be applicable to multi-step reasoning. Therefore, investigating the mechanisms of multi-step reasoning is an important and meaningful topic.
> > > > >
> > > > > (3)As shown in the appendix E (parallel reasoning), when the reasoning step count $n \geq$ the number of model layers $L$, the model exhibits parallel reasoning behavior. This further demonstrates that "linearly stacking multiple single-step reasoning processes" cannot describe all mechanisms used by models to achieve multi-step reasoning.
> > > > >
> > > > > [Application to Phi-3 Model]
> > > > >
> > > > > Thank you for your insightful suggestions. In Appendix H of the revised manuscript, we include relevant experiments. Contrary to expectations, some heads consistently exhibit diagonal patterns across layers in large models, leading to diagonalized $W^{vo(l)}$ and $W^{qk(l)}$ from a layer-wise perspective. However, we designed two experiments to demonstrate that **the diagonal patterns in $W^{qk}W^{vo,T}$ are not entirely caused by the diagonalization of $W^{qk}$ and $W^{vo}$ themselves but are more often due to weight alignment.**
> > > > >
> > > > > (1)Setting the diagonal elements of $W^{qk}$ and $W^{vo}$ to 0 shows that the kernel score (KS) of $W^{qk}W^{vo,T}$ remains largely unchanged (Figure 21(f)).
> > > > >
> > > > > (2)Using noise-added identity matrices $A=I+aN(0,0.1)$ and $B=I+aN(0,0.1)$, we evaluated their kernel score (KS). When $\max(\text{KS}(A), \text{KS}(B))<0.3$, $\text{KS}(C)<0.025$ for $C=AB$ (Figure 21(d)). However, in Phi-3, we found many cases where $\text{max}(\text{KS}(W^{qk(l_1)}), \text{KS}(W^{vo(l_2),T}))<0.3$ but $\text{KS}(W^{qk(l_1)}W^{vo(l_2),T})>0.3$ (Figure 21(c)). This indicates that weight alignment significantly impacts the formation of diagonal structures.
> > > > >
> > > > > The primary purpose of including the Phi-3 model section is to address Reviewer [JU6V]’s question: “Would it be reasonable for block-diagonal weight?” We demonstrate that weight alignment phenomena also occur in real large-scale language models. Your suggestion has helped make our explanation more comprehensive and rigorous.
> > > > >
> > > > > Another function of this section is to extend the matching/kernel score discussed in the main text to the multi-head model scenario. The score, as defined in our framework, effectively reflects the location and characteristics of reasoning heads in the model, aligning well with conclusions from some previous studies.
> > > > >
> > > > > However, this example alone cannot strictly and directly support the claim:
> > > > > >The buffer mechanism can also be observed in real language models.
> > > > >
> > > > > In the revised manuscript, we have removed this overclaim.
> > > > >
> > > > > [Transformer Block Coupling]
> > > > >
> > > > > (Aubry, Murdock, et al. 2024) is an intriguing parallel work that identifies weight alignment phenomena in large models through Jacobian matrices. We will cite this work in the revised manuscript and follow up on its developments in future research.
> > > > >
> > > > > **We sincerely appreciate your thorough and patient review of our manuscript. Please do not hesitate to reach out if you have any further questions or require additional clarifications!**

---

> ### Comment · Reviewer_fnjG · 2024-11-26
>
> Thanks for providing the additional experiments and clarifications. I have increased my score to 5 to reflect my greater confidence in the soundness of the experiments.
>
> However, after the authors' clarification that they do not in fact have solid evidence for the buffer mechanism in Phi-3, I'm slightly concerned about the applicability of the results.
>
> Specifically, it's possible that the buffer mechanism is a product of the particular training set up you used, and they may not emerge in pretrained language models. One reason why this might be true is that pretrained LMs tend to find directions in activation space that correspond to semantic concepts. This view is fundamentally at odds with the buffer mechanism, which says instead that there are multiple replicas of a semantic space.
>
> Are there more concrete ways of determining if the buffer mechanism is used in Phi-3? This is the crux for me that will raise my score to a 6 or an 8.
>
> A minor point: Sec. J is very sparse on details. In particular, Fig. 24 should probably report quantitative metrics for various predictions, and also some _sense_ of the statistical significance. For example, you might report how frequently are the interventions working as your claim, and how many data points did you test with.

---

> > ### Author Response · Authors · 2024-11-26
> >
> > Thank you very much for your recognition of our work and for the further questions you raised. We address the issues related to the Phi-3 model that you mentioned as follows:
> >
> > (1)Firstly, the removal of the absolute phrasing in the original text was not render an information that the "buffer mechanism is not in fact observed in Phi-3." On the contrary, **our numerous experiments provide substantial evidence that the buffer mechanism highly possibly is used in real large models**:
> >
> > (a)We defined the kernel/matching score of the multi-head model, and the computed scores align well with the positions and characteristics of the inference heads in the model, consistent with findings from previous studies.
> >
> > (b)In the simplest form of multi-step reasoning, i.e., single-step reasoning, we demonstrated that the weight alignment phenomenon mentioned in buffer theory is real and observable.
> >
> > (c)**(new)** Another strong evidence supporting the existence of the buffer mechanism is the computation of cosine similarity between {$W^{vo(l)}$}’s. We have added an experiment in the revised manuscript to compute cosine similarity, and the result is that $cos sim(W^{vo(l_1)},W^{vo(l_2)}) \approx 0$ when$l_1 \neq l_2$.
> >
> > These experiments indirectly suggest that the buffer mechanism is likely adopted in real large models. However, as you and other reviewer pointed out earlier, it is nearly impossible to exclude all other potential mechanisms in highly complex real-world large language models. **We removed the absolute phrasing just for the sake of rigor.** The following content is added in the main text:
> >
> > > Phenomena such as weight alignment and $W^{vo(l)}$ diversity provide evidence for the presence of the buffer mechanism in real language models.
> >
> > (2)As suggested by you and other reviewers, the primary aim of this paper is to propose, **through experiments and observations on simple tasks, potential mechanisms that may be employed by Transformer models when performing multi-step reasoning. This helps to shed light on the advantages of the Transformer architecture.**
> >
> > (3)Our work provides **heuristic insights** into how the reasoning capabilities of large language models can be enhanced in the future. For example, the **RMBA**, designed based on the buffer mechanism, significantly improved model performance on the PrOntoQA dataset. Future work could explore **incorporating "qk & vo alignment" or "vo diversity" into the loss function** during training to further boost model reasoning abilities. Alternatively, we consider **developing causal intervention experiments at the buffer level**, as shown in Appendix J, to more precisely probe useful information.
> >
> > Regarding the **causal intervention** experiments, we have provided a detailed flowchart in the appendix and added the following example to explain how causal intervention can be performed:
> >
> > > For example, as shown in Fig.23(right), suppose we want to change the information [a] stored in the buffer $W^{vo(0)}W^{vo(1)}$ in layer 2 to [x]. This can be achieved by simply modifying the input sentence [a’][b’][b’][c’][a][b][b][c]...[a]to [a’][b’][b’][c’][x][b][b][c]...[a], and enforcing $Attn^{(1)}{[13:]}$ same as before.
> >
> > For each scenario, we traversed [40,100] for [x], conducting at least **600 experiments in total** and obtain consistent results, that is, for cases where the table does not indicate "Random," **the probability of the model output deviating from the value presented in the table is less than 1e−15**. We report this result in the caption as follows:
> >
> > > In this experiment, the tokens in the original sentence are selected from the range [20, 40], while token [x] traverses the range [40, 100]. $Attn^{(l)}{[13:]}$ refers to the attention score corresponding to the last token at layer $l$. For the original input, we have $ Attn^{(1)}{[13,6]} = 1$ and $ Attn^{(2)}{[13,8]} = 1$. Instances labeled as "Random" indicate that the output varies erratically as [x] changes. In all other cases, the probability of the model output deviating from the value presented in the table is less than 1e−15.
> >
> > The corresponding third-round modifications are highlighted in the revised manuscript using green text. **We sincerely appreciate the valuable feedback provided by the reviewer and deeply value your thoughtful insights. We also hope to continue receiving your support in advancing this original work. Should you have any further questions or suggestions, please feel free to reach out to us at any time. We would be happy to continue the discussion.**

---

> > ### Author Response · Authors · 2024-12-02
> >
> > Dear Reviewer, we would like to thank you again for your valuable feedback during the review process. As the discussion phase is nearing its conclusion, this message is to confirm whether you have seen our previous response (about phi-3 and causal intervention) and if you are satisfied with the updates we provided.
> >
> > In our last response, we included several new pieces of evidence supporting the existence of the buffer mechanism in the Phi-3 model. Additionally, we acknowledge your concerns regarding the semantic space in real language models. In fact, previous works have shown that the semantic space of large models is often a low-dimensional manifold. For example, through 2D PCA analysis, we can observe clustering structures within the embedding matrix. However, generally speaking, both token embeddings and hidden spaces in large language models are high-dimensional vectors. This suggests that in high-dimensional vector spaces, it is natural for large models to employ buffers to store different attributes of an entity, thus creating multiple copies of semantic spaces. This ensures that the sum of two semantic vectors will not lead to a loss of the original meaning. A simple analogy is as follows: if the sum of two positive integers is known to be 5, we cannot determine whether the addends are (2,3) or (1,4). However, if the sum of two 2-dimensional addends, where only one element is non-zero, is [1,4], we can easily determine that the addends are [1,0] and [0,4], i.e., [1,4] = 1*[1,0] + 4*[0,1] (here, the two buffers are [1,0] and [0,1]).
> >
> > **We sincerely appreciate your contributions to this work over this period, and we hope to receive your continued support.**

---

> ### Comment · Reviewer_fnjG · 2024-12-03
>
> Thanks for adding the experiments with the cosine similarity on $W^{ov}$ matrices for Phi-3.
>
> I still believe that this line of weight alignment argument is insufficient for establishing that Phi-3 exhibits the buffer mechanism. The weight alignment phenomena you're highlighting in Sec. H seems extremely general because it's a property of the Phi-3 weights, and not a property of any particular symbolic reasoning task you're studying for the main paper. This makes the hypothesis space much larger, because there could be many training dynamics considerations that could lead the formation of this weight alignment pattern.
>
> If you can propose a concrete task on which Phi-3 exhibits the buffer mechanism, it will greatly improve the relevance and impact of the paper. Such evidence can come in the form of analyzing the representational role of activations, or analyzing behaviors of attention heads in this concrete task. I believe that if this is rigorously shown in an LLM, this paper can be really important.
>
> Overall, while I appreciate the authors' efforts at improving the experimental rigor, I still believe that the the paper's contributions are limited because the authors made the choice to focus their analysis on synthetically trained models on toy synthetic datasets, rather than an LLM. I will therefore choose not to increase my score.

---

> > ### Author Response · Authors · 2024-12-03
> >
> > Dear Reviewer,
> >
> > First and foremost, we would like to express our sincere gratitude for the substantial and constructive feedback you have provided on our work. Your insightful suggestions have greatly contributed to the overall improvement of our manuscript. We highly appreciate your decision to revise your initial evaluation, and we respect your final assessment of our paper.
> >
> > Many of your suggestions have been particularly thought-provoking. For instance, the mention of causal intervention led us to consider the potential development of a buffer-level causal intervention method (similar to Appendix J) in future work. This approach could allow for better identification of key information and further support the existence of a buffer mechanism within large models. While we acknowledge your concern that the three pieces of evidence we provided may not be directly conclusive, we still believe that there is no need for undue skepticism regarding the presence of the buffer mechanism in large language models. At the very least, our work demonstrates that the Transformer architecture can leverage buffer mechanisms to process diverse information, showcasing its architectural advantages. In future research, we will explore how to identify more concrete examples for the direct validation of the buffer mechanism’s existence and aim to develop more refined causal intervention methods.
> >
> > Once again, we sincerely appreciate your positive and constructive feedback throughout this process. We wish you all the best in your future endeavors.

---

### Author Response · Authors · 2024-11-18
**Common Response**

We would like to express our sincere gratitude to the reviewers for their valuable insights and comments, which we have incorporated into our revised manuscript. In response to the reviewers' suggestions, we have devoted considerable time and effort to implement several significant revisions, including the addition of supplementary experimental results and numerous improvements to the overall presentation of the paper. Below are the specific modifications and additions we have made. **The changes in the revised manuscript are marked in red.**

**[R1][Presentation]** We have meticulously refined the captions of every figure and table in the manuscript. The derivations of the matching matrix and kernel matrix, which are crucial to our paper, have been rewritten to eliminate any potential ambiguities present in the original version. Additionally, we have revised some statements in the original text that might have appeared as overclaims.

**[R2][Definition of Formulas]** we have **redefined the kernel score** utilizing the normalization property of the softmax function. The new kernel score now ranges within [0, 1], making it more interpretable.

**[R3][Supplementary Experiments]** According to the reviewers' suggestions and to enhance the persuasiveness of the paper, we have added the following experimental content:

- **Hyperparameter Sweep for RMBA and IMBA Algorithms.** In the appendix of the new manuscript, we have included results demonstrating the impact of different hyperparameters on the outcomes of the three methods. We selected 4 weight decay values, 5 learning rates, and 4 configurations of hidden space dimensions. Under each setting, we conducted 3 experiments for each of the three algorithms (baseline, IMBA, RMBA), **totaling 720 experiments**. In most cases, RMBA achieved better results than the other algorithms. Additionally, we have supplemented the loss curves of the three algorithms on the PrOntoQA task, showing that in terms of training stability, RMBA is comparable to the baseline.

- **Training a 2-Layer Transformer with only single-Step Reasoning Data to Perform multi-Step Reasoning via CoT.** In the appendix, we demonstrate how a 2-layer Transformer network, trained with only single-step reasoning data, can use Chain-of-Thought (CoT) to execute multi-step reasoning tasks. We have provided detailed information flow diagrams, rather than the schematic diagrams presented in the main text.

- **Calculating Matching Score and Kernel Score in the Phi-3 Model.** We employed the open-source Phi-3 large language model and added methods for calculating the matching score and kernel score in multi-head models. We computed the strength of information interaction between layers of Phi-3 using the kernel score and calculated the matching score of each head in each layer. We found:

  - **(a)** When $l_1 \leq l_2$, $W^{qk(l_1)} W^{vo(l_2),T}$ has almost no significant features (kernel score ≈ 0).

  - **(b)** When $l_1 > l_2$, the kernel score of $W^{qk(l_1)} W^{vo(l_2),T}$ decreases as $l_1 - l_2$ increases. (a)(b) indicate that information stored in earlier layers (with buffer $W^{vo(l_2)}$) is extracted by later layers (via $W^{qk(l_1)}$). Moreover, the model tends to extract the most recently processed information.

  - **(c)** Heads with high matching scores are mostly concentrated in the middle layers. This aligns with previous research findings that reasoning in large models predominantly occurs in middle layers.

- **Conducting More Rounds of Interaction with Large Language Models to Evaluate Their Reasoning Abilities.** We used 4 GPT and Claude models to perform 2-step, 3-step, and 4-step reasoning tasks, examining the models' performance both when allowed to use CoT and when not allowed. For each setting, we repeated the interaction 25 times, **totaling 600 interactions**, to provide statistically significant results. The results indicate that existing large models find it challenging to directly perform 2-step or more reasoning tasks effectively. And with the aid of CoT, these models can excel in multi-step reasoning tasks.

---

### Meta-Review · Area_Chair_zSuH · 2024-12-23

**Metareview:**

The paper introduces a buffer mechanism in transformer models for multi-step reasoning tasks and proposes a Random Matrix-Based Algorithm (RMBA) to improve training efficiency. While the study offers insights into reasoning processes in transformers, it has significant weaknesses. The reliance on synthetic tasks and simplified datasets limits the generalizability of findings to real-world large models. Experimental evidence for the buffer mechanism in practical language models remains insufficient, and key claims about the mechanism’s role are overly assertive without rigorous empirical validation. The evaluation lacks direct comparisons to alternative hypotheses, such as overwrite mechanisms, reducing the robustness of the conclusions.

Strengths include addressing an important problem, novel theoretical insights, and detailed experimental analysis on synthetic datasets. However, these are outweighed by the lack of generalizability, insufficient experimental rigor, and reliance on synthetic benchmarks.

**Additional Comments On Reviewer Discussion:**

The rebuttal addressed some concerns about clarity and added experiments on the Phi-3 model, but it failed to provide compelling evidence of the buffer mechanism’s existence in real-world settings. Reviewers were unconvinced by the generalizability of the findings and highlighted the limited scope of experiments. Claims about CoT reasoning and buffer mechanisms remain speculative without concrete empirical support. While the authors made substantial revisions and improved the clarity of the manuscript, the fundamental limitations in scope and validation justify the rejection decision.

---

### Decision · Program_Chairs · 2025-01-22

Reject